# Targeting Mitochondrial COX-2 Enhances Chemosensitivity via Drp1-Dependent Remodeling of Mitochondrial Dynamics in Hepatocellular Carcinoma

**DOI:** 10.3390/cancers14030821

**Published:** 2022-02-06

**Authors:** Lin Che, Jia-Shen Wu, Ze-Bang Du, Yu-Qiao He, Lei Yang, Jin-Xian Lin, Zhao Lei, Xiao-Xuan Chen, Dong-Bei Guo, Wen-Gang Li, Yu-Chun Lin, Zhong-Ning Lin

**Affiliations:** 1State Key Laboratory of Molecular Vaccinology and Molecular Diagnostics, School of Public Health, Xiamen University, Xiamen 361102, China; 32620170154849@stu.xmu.edu.cn (L.C.); 32620210156090@stu.xmu.edu.cn (J.-S.W.); 32620191150574@stu.xmu.edu.cn (Z.-B.D.); 32620201150808@stu.xmu.edu.cn (Y.-Q.H.); 32620201150760@stu.xmu.edu.cn (L.Y.); 32620191150579@stu.xmu.edu.cn (J.-X.L.); leizhao@xmu.edu.cn (Z.L.); chen_xuan17@xmu.edu.cn (X.-X.C.); guodb@xmu.edu.cn (D.-B.G.); 2Department of Hepatobiliary Surgery and Pancreatic & Organ Transplantation Surgery, Xiang’an Hospital of Xiamen University, School of Medicine, Xiamen University, Xiamen 361102, China; lwgang@xmu.edu.cn; 3Cancer Research Center, School of Medicine, Xiamen University, Xiamen 361102, China

**Keywords:** mitochondrial cyclooxygenase-2, dynamin-related protein 1, mitochondrial dynamics, sirtuin 3, apoptosis, hepatocellular carcinoma

## Abstract

**Simple Summary:**

New therapeutic strategies are urgently needed to improve the anti-cancer effect for hepatocellular carcinoma (HCC). Overexpression of cyclooxygenase-2 (COX-2) is found in several types of cancers and correlates with a poor prognosis. However, it remains unclear how the mitochondrial translocation of COX-2 is involved in mitochondrial dynamics and sensitizes HCC cells to multipattern anti-tumor therapy. We explored the impact of targeting mitochondrial COX-2 (mito-COX-2) intervention toward mitochondrial dynamics on platinum-based chemotherapeutics in HCC cells and xenograft nude mouse models. Our study indicates that the mito-COX-2 represents a candidate predictive biomarker and potential target to regulate anti-cancer sensitization of HCC, and possibly for other types of COX-2-high-expression cancers.

**Abstract:**

Mitochondria are highly dynamic organelles and undergo constant fission and fusion, which are both essential for the maintenance of cell physiological functions. Dysregulation of dynamin-related protein 1 (Drp1)-dependent mitochondrial dynamics is associated with tumorigenesis and the chemotherapeutic response in hepatocellular carcinoma (HCC). The enzyme cyclooxygenase-2 (COX-2) is overexpressed in most cancer types and correlates with a poor prognosis. However, the roles played by the translocation of mitochondrial COX-2 (mito-COX-2) and the interaction between mito-COX-2 and Drp1 in chemotherapeutic responses remain to be elucidated in the context of HCC. Bioinformatics analysis, paired HCC patient specimens, xenograft nude mice, immunofluorescence, transmission electron microscopy, molecular docking, CRISPR/Cas9 gene editing, proximity ligation assay, cytoplasmic and mitochondrial fractions, mitochondrial immunoprecipitation assay, and flow cytometry analysis were performed to evaluate the underlying mechanism of how mito-COX-2 and p-Drp1^Ser616^ interaction regulates the chemotherapeutic response via mitochondrial dynamics in vitro and in vivo. We found that COX-2 and Drp1 were frequently upregulated and confer a poor prognosis in HCC. We also found that the proportion of mito-COX-2 and p-Drp1^Ser616^ was increased in HCC cell lines. In vitro, we demonstrated that the enhanced mitochondrial translocation of COX-2 promotes its interaction with p-Drp1^Ser616^ via PTEN-induced putative kinase 1 (PINK1)-mediated Drp1 phosphorylation activation. This increase was associated with higher colony formation, cell proliferation, and mitochondrial fission. These findings were confirmed by knocking down COX-2 in HCC cells using CRISPR/Cas9 technology. Furthermore, inhibition of Drp1 using pharmacologic inhibitors (Mdivi-1) or RNA interference (si*DNM1L*) decreased mito-COX-2/p-Drp1^Ser616^ interaction-mediated mitochondrial fission, and increased apoptosis in HCC cells treated with platinum drugs. Moreover, inhibiting mito-COX-2 acetylation with the natural phytochemical resveratrol resulted in reducing cell proliferation and mitochondrial fission, occurring through upregulation of mitochondrial deacetylase sirtuin 3 (SIRT3), which, in turn, increased the chemosensitivity of HCC to platinum drugs in vitro and in vivo. Our results suggest that targeting interventions to PINK1-mediated mito-COX-2/p-Drp1^Ser616^-dependent mitochondrial dynamics increases the chemosensitivity of HCC and might help us to understand how to use the SIRT3-modulated mito-COX-2/p-Drp1^Ser616^ signaling axis to develop an effective clinical intervention in hepatocarcinogenesis.

## 1. Introduction

Hepatocellular carcinoma (HCC) ranks as the second cause of cancer-related deaths worldwide [1]. Major risk factors, including infection with hepatitis B and C viruses, alcohol consumption, aflatoxin contamination, and selective expression of oncogenes, contribute significantly to hepatocarcinogenesis [2,3,4]. Although the patterns of gene-environment interactions in HCC predisposition have been gradually uncovered and treatments have improved over the last decades, the prognosis is still poor, due to late diagnosis and low response to therapies, mainly linked to chemoresistance [5,6]. Therefore, it is urgent to discover new therapeutic strategies targeting oncogenic signaling pathways for the development of next-generation HCC therapeutics.

Mitochondria form dynamic organelle networks in the cell that are balanced via the molecular events of alternating fusion and fission [7]. Mitochondria are amplifiers of ATP production, redox signaling, and metabolic functions in tumors. It is well-known that mitochondrial dysfunction plays a crucial role in tumor transformation. As the cell powerhouse, the mitochondrial dynamics have been identified as a key therapeutic target for the treatment of cancer [8]. Both processes are essential for the regulation of cellular homeostasis, including oxidative stress, signal transduction, metabolism, and cell apoptosis susceptibility [9,10,11]. During the initiation, growth, and survival of tumor cells, significant mitochondrial function changes take place in the important enzymes of molecular signaling, the tricarboxylic acid cycle, and mitochondrial dynamics. Therefore, enzymes regulating mitochondrial dynamic networks’ homeostasis are key mediators of tumorigenesis [12]. Dynamin-related protein 1 (Drp1), a member of the dynamin family of GTPases, is the key component of the mitochondrial fission machinery regulating mitochondrial quality control (MQC). Recently, our studies have shown that Drp1 mediates necroptosis in hepatocytes and hepatic injury induced by cadmium. However, it remains unclear whether the activation of Drp1 phosphorylation affects its mitochondrial translocation and regulation of hepatic injury [13]. Phosphorylation is one of the best characterized post-translational modifications (PTMs) of Drp1, which can unleash both activating and repressing effects dependent on the specific functional site(s) of phosphorylation. It was reported that inactive Drp1 with phosphorylation at serine 637 (p-Drp1^Ser637^) shifts the mitochondrial balance toward fusion. Conversely, activated Drp1 with phosphorylation at serine 616 (p-Drp1^Ser616^) translocates to mitochondria, where it binds protein partners located on the outer mitochondrial membrane (OMM) and subsequently promotes mitochondrial fission [14]. Emerging evidence has revealed that mitochondrial fission was involved in the inhibition of multiple patterns of cell death such as apoptosis and necroptosis [15,16]. Thus, not surprisingly, mitochondria are important mediators of tumorigenesis. Excessive mitochondrial fission has been associated with a poor prognosis in HCC patients [17]. Recently, reports have shown that overexpression of p-Drp1^Ser616^ was involved in chemoresistance of several human tumor types, including HCC [18,19]. Thus, understanding the mechanisms underlying p-Drp1^Ser616^-dependent mitochondrial regulation during hepatocarcinogenesis and translating these discoveries into specific therapeutic targets are critical to decision-making and improving patients’ responses to therapy and prognoses.

Cyclooxygenases (COXs), including COX-1 and COX-2, are key enzymes for arachidonic acid conversion into prostaglandins [20]. Encoded by the prostaglandin-endoperoxide synthase 2 (*PTGS2*) gene, inducible COX-2 is normally absent from most cells but can be induced in malignantly transformed cells in response to oncogenic stress [21,22]. COX-2 is localized in the endoplasmic reticulum (ER) and is considered an ER-retained protein. Recently, ours and other studies have shown that COX-2 is also present in mitochondria [23,24]. Mitochondrial COX-2 (mito-COX-2) has been recognized as a potential theranostic target against cancer stem cells in nasopharyngeal carcinoma [24]. In addition, our research found that liver carcinogenic mycotoxin aflatoxin B1 combined with Hepatitis B virus x protein triggers hepatic steatosis via COX-2-mediated mitochondrial dynamics disorder [25]. However, it remains unclear how the mitochondrial translocation of cytoplasmic COX-2 is induced, and whether mito-COX-2 participates in the p-Drp1^Ser616^-dependent mitochondrial dynamics disorder and leads to carcinogenesis in the liver.

Interestingly, COX-2 contains phosphorylation and acetylation sites, which the PTMs of COX-2 can use to regulate its function and for subcellular localization. Our previous study showed that COX-2 phosphorylated at serine 601 (p-COX-2^Ser601^) undergoes ER retention for proteasome degradation [26]. However, the mechanism by which the PTM of COX-2 acts on its mitochondrial distribution and function remains to be elucidated. It is known that the key proteins required for mitochondrial dynamics are regulated by their acetylation state, as acetylation promotes mitochondrial fission. Therefore, the mechanism by which the mito-COX-2 acetylation status affects its mitochondrial distribution and function has attracted our attention. Sirtuin 3 (SIRT3), the major deacetylase in mitochondria, is involved in the regulation of mitochondrial homeostasis by offering protection from a variety causes of damage. SIRT3 is widely expressed in mitochondria-rich tissues such as the liver [27]. Given the crucial role of mitochondria in metabolism, intracellular signaling, and apoptosis, highly metabolic liver tissues are particularly sensitive to mitochondrial dysfunction. A growing body of evidence shows that SIRT3 inhibits tumorigenesis by deacetylation of its substrates in HCC mitochondria, pointing to SIRT3 as a potential therapeutic target [28]. It was found that SIRT3 acts as a tumor suppressor in HCC by inducing apoptosis through the glycogen synthase kinase-3β (GSK-3β)/Bax signaling pathway [28]. SIRT3 plays a critical role in repairing mitochondrial DNA damage and preventing apoptosis under oxidative stress by deacetylating human 8-oxoguanine-DNA glycosylase 1 (OGG1) [29]. However, no one has yet explored whether SIRT3 regulates deacetylation associated with mito-COX-2-mediated mitochondrial dynamics in the context of HCC.

Platinum-based antitumor agents, including cisplatin (cDDP), carboplatin (CBP), and oxaliplatin (L-OHP), are active against many tumor types. All three drugs induced platinum-mediated DNA damage [30]. Research has shown that cDDP may directly interact with mitochondria to induce ovarian cancer cells’ apoptosis, which may account for our understanding of the clinical activity of cDDP and development of resistance [31]. However, the basis for the roles of mitochondria under treatment with chemotherapy is poorly understood. In the present study, we aimed to investigate the role of the SIRT3-modulated mito-COX-2/p-Drp1^Ser616^ axis in mitochondrial dynamics and apoptosis of HCC cells, and elucidate the mechanisms of PINK1-dependent mito-COX-2/p-Drp1^Ser616^ interaction implicating the mitochondrial protein machinery. We explored the impact of targeted mito-COX-2 intervention toward mitochondrial dynamics on platinum-based chemotherapeutics in HCC cells and xenograft nude mouse models. Our study suggests that mito-COX-2 is a theranostic biomarker and potential target to regulate multi-pattern anti-cancer sensitization of HCC.

## 2. Materials and Methods

### 2.1. Bioinformatics Analysis

Two publicly available mRNA expression datasets (GSE104310 and GSE36376) from the Gene Expression Omnibus (GEO) database (http://www.ncbi.nlm.nih.gov/geo/, access on 8 January 2020) were screened for analysis. Detailed information is presented in Appendix A. Standardization and expression value calculation were performed on the original datasets using the Impute and Limma packages of R software (version: x64 4.0.2). The fold-change (FC) and p-values were used to screen differentially expressed genes (DEGs). Heatmap and volcano plots of DEGs were constructed, using Pheatmap (v1.0.12), Ggplot2 (v3.3.5), and other software packages. Gene ontology (GO) analysis and Kyoto Encyclopedia of Genes and Genomes (KEGG) pathway analysis were used to identify pathways significantly affected by the hepatocarcinogenesis process of DEGs. Single-gene set enrichment analysis (GSEA) was used to determine various signaling signatures related to the expression of *PTGS2* and *DNM1L* genes, and the pathways with a false discovery rate (FDR) < 0.25 were considered significantly enriched. Protein-protein interaction (PPI) networks analysis was performed to identify mitochondrial dynamics-related proteins likely to interact with COX-2 using the online STRING database (https://string-db.org/, access on 6 March 2020), and results of the analysis were imported into Cytoscape 3.7.1 to establish a network model. Kaplan-Meier curve analysis of the overall survival in HCC patients was performed using the UALCAN database (http://ualcan.path.uab.edu/, access on 11 January 2020).

### 2.2. HCC Tissue Specimens

Twelve human HCC tumor tissues and their corresponding peritumoral tissues were obtained from patients who underwent liver tumor resection at Xiang’an Hospital affiliated with Xiamen University (Xiamen, China). All the tissue specimens were immediately frozen in liquid nitrogen and then stored at −80 °C before use. Some of these specimens were fixed and paraffin-embedded before histological examination. The levels of the indicated proteins (COX-2, p-Drp1^Ser616^, SIRT3, and so on) were detected by western blotting (WB), immunohistochemical (IHC) staining, and immunofluorescence (IF) assay. All HCC patients involved in this study gave written informed consent and were approved by the Ethics Committee of Xiang’an Hospital affiliated with Xiamen University.

### 2.3. In Vivo Subcutaneous Xenograft Models

Six-week-old BALB/c nude mice were purchased from SLAC Laboratory Animal Co., Ltd. (Shanghai, China). All experiments were approved by the Experimental Animal Ethics Committee of Xiamen University (ethics protocol code: XMULAC20180094). To explore the effect of COX-2-overexpression and COX-2-knockdown on the growth of HCC cells’ xenograft tumors, mice were subcutaneously inoculated with HepG2-pB-MTS-Flag or HepG2-pB-MTS-*PTGS2*-Flag cells, and HepG2-Cas9-NC or HepG2-Cas9-*PTGS2* cells, in the flanks, respectively. In the intervention model, at day 10 after inoculation, the mice were randomly divided into four treatment groups (*n* = 5 for each group) each with a different administration: normal saline, cisplatin (cDDP), resveratrol (RSV), and cDDP + RSV. The drugs were injected subcutaneously and the tumor volume was recorded every three days for three weeks. The mice were sacrificed to observe the size and measure the weight of the excised xenograft tumors. Xenograft tumor tissues were used for histological examination, mitochondrial and cytoplasmic fractions’ preparation, and measurement of proteins’ expression levels by WB and IF assay.

### 2.4. Immunohistochemical (IHC) Analysis

The tissue sections from samples of HCC patients and xenograft tumors of BALB/c nude mice were fixed with 4% paraformaldehyde (PFA), embedded with paraffin, and serially sectioned at 4 μm thickness. IHC detection was performed using an UltraSensitive^TM^ SP IHC Kit (MXB, Fuzhou, China) as described in our previous study [32], with the primary antibodies being anti-COX-2, anti-Drp1, anti-p-Drp1^Ser616^, or anti-SIRT3. Quantitative analysis was performed with Image Pro-Plus software 6.0 (IPP 6.0, Media Cybernetics, Rockville, MD, USA).

### 2.5. Cell Culture

Immortalized human liver cell line L02, HCC cell lines HepG2, MHCC97H, and Huh7, and human embryonic kidney 293T (HEK-293T) cells were obtained from the Cancer Center of Sun Yat-sen University (Guangzhou, China). Human hepatic cell line QSG7701 and HCC cell line QGY7703 were obtained from the College of Chemistry and Chemical Engineering, Xiamen University (Xiamen, China). Cells were cultured following routine protocol as previously described [32].

### 2.6. Establishment of Stable Cell Lines and RNA Interference

To establish stable COX-2-knockdown cell lines, small guide RNA (sgRNA)-coding cDNAs targeting the *PTGS2* gene were designed and synthesized for the construction of Lenti-Cas9-*PTGS2* recombinant plasmids. The *PTGS2*-sgRNAs were annealed and cloned into the LentiCRISPRv2 vector (Addgene, Watertown, MA, USA). The primer sequences were as followings: *PTGS2*-sgRNA-1, forward primer (FP): 5′-CACCGAACTCATAATTGCATTTCGA-3′, reverse primer (RP): 5′-AAACTCGAAATGCAATTATGAGTTC-3′. *PTGS2*-sgRNA-2, FP: 5′-CACCGCGTTCCAAAATCCCTTGAAG-3′ and RP: 5′-AAACCTTCAAGGGATTTTGGAACGC-3′. Lenti-Cas9-*PTGS2*-expressing constructs or the negative control (NC) plasmids Lenti-Cas9-NC were transfected to HEK-293T cells using the Lipofectamine 2000 reagent (Invitrogen, Carlsbad, CA, USA), according to the manufacturer’s instructions. The transfected cells were selected with 0.6 μg/mL puromycin to generate a polyclone of cells with stable knockdown of COX-2. They were HepG2-Cas9-*PTGS2* and MHCC97H-Cas9-*PTGS2* cells for COX-2-knockdown cells, and their controls were HepG2-Cas9-NC and MHCC97H-Cas9-NC cells.

Construction of stable COX-2-overexpressing cell lines was carried out as previously described [24]. The HepG2-*PTGS2* and HepG2-MTS-*PTGS2*-Flag (MTS: mitochondrial targeting sequence) cells, and MHCC97H-*PTGS2* and MHCC97H-MTS-*PTGS2*-Flag cells, were the COX-2- and mito-COX-2-overexpressing cell lines, while the HepG2-pB-MTS-Flag cells and MHCC97H-pB-MTS-Flag cells were the corresponding control cells transfected with an empty vector. The HepG2- and MHCC97H-*SIRT3* cells were the stable SIRT3-overexpressing cell lines, while HepG2- and MHCC97H-pBabe cells were the corresponding control cells.

For transient transfection, cells were seeded in six-well plates to 70% confluence. Small interfering RNAs (siRNAs) for the *DNM1L* gene (si*DNM1L*, for knockdown of Drp1), and their corresponding negative controls (siNCs) were synthesized by Ribobio Co. (Guangzhou, China). The siRNAs were transfected into cells using the Lipofectamine 2000 reagent (Invitrogen) according to the manufacturer’s instructions.

### 2.7. Antibodies and Reagents

The following antibodies were used in western blotting assays: anti-COX-2, anti-AKT, anti-ERK, anti-CDK2, anti-Ac-lysine, and anti-SIRT3 (Cell Signaling Technology, Danvers, MA, USA), anti-Drp1 and anti-p-Drp1^Ser616^ (Ruiying Biological, Jiangsu, China), anti-p38 MAPK, anti-PKCα, anti-PINK1, anti-GAPDH, anti-Bax, anti-Bcl-2, anti-cleaved-Caspase-3, anti-COXIV, anti-Flag, Alexa Fluor^®^ 488-labeled goat anti-rabbit IgG, Alexa Fluor^®^ 488-labeled goat anti-mouse IgG, and DyLight^®^ 405-labeled goat anti-mouse IgG (Beyotime, Shanghai, China). The peroxidase-conjugated affinipure secondary antibodies goat anti-rabbit IgG and anti-mouse IgG were purchased from Thermo Fisher Scientific (Waltham, MA, USA). cDDP, RSV, dimethyl sulfoxide (DMSO), and Mdivi-1 were purchased from Sigma Aldrich (St. Louis, MO, USA). Carboplatin (CBP) was purchased from Hansoh Pharmaceutical Co., Ltd. (Jiangsu, China), and oxaliplatin (L-OHP) was purchased from Selleck (Santa Clara, TX, USA).

### 2.8. Western Blotting Analysis

Western blotting analysis was performed as described previously [32]. Briefly, the cells, HCC tumor and peritumoral tissues, and xenograft tumor tissues were lysed in whole-cell lysate buffer or radio-immunoprecipitation assay (RIPA) buffer with a 1% phosphatase inhibitor cocktail (Beyotime, Shanghai, China). The indicated protein bands were incubated overnight at 4 °C with primary antibodies of the target proteins (such as anti-COX-2, anti-Drp1, anti-p-Drp1^Ser616^, anti-PINK1, anti-SIRT3 antibodies, etc.), and then incubated with the corresponding secondary antibodies at room temperature for 1 h. Signals were visualized using the Azure Biosystems (Beijing, China).

### 2.9. Mitochondrial and Cytoplasmic Fractions and Mitochondrial Protein Immunoprecipitation (Mito-IP)

Mitochondrial and cytoplasmic fractions were obtained as described previously using a mitochondrial isolation kit (Enzo Life, New York, PA, USA) [13]. The levels of COX-2, Drp1, p-Drp1^Ser616^, PINK1, and SIRT3 proteins in both mitochondrial and cytoplasmic fractions were detected by WB analysis. COXIV and GAPDH were used as loading controls to monitor the mitochondria and cytoplasm, respectively. Mito-IP analysis was performed as described previously [13]. Briefly, SureBeads (100 μL) (BioRad, Hercules, CA, USA) were washed with 1× PBST (PBS plus 0.1% Tween 20) three times and supernatant was discarded using magnetic separator. Then, 200 μL primary antibodies (anti-COX-2, anti-p-Drp1^Ser616^, anti-Flag, or anti-SIRT3) were adhered to SureBeads in a rotary rocking bed for 20 min. Then, a mitochondrial fraction was adhered to the beads at room temperature for 1 h. The beads were washed three times with 1× PBST and the proteins were eluted with 1× SDS buffer and analyzed using routine WB.

### 2.10. Immunofluorescence (IF) Assay

To study the co-localization of mitochondria and COX-2, p-Drp1^Ser616^, or SIRT3, cells were mounted on coverslips and stained with 100 nM MitoTracker Red CMXRos (Invitrogen) for 30 min. IF assay was performed as described previously [13]. Briefly, after being fixed, permeabilized, and blocked, the cells were incubated overnight with primary antibodies (anti-COX-2, anti-p-Drp1^Ser616^, or anti-SIRT3), followed by the appropriate fluorescent secondary antibodies for 1 h in the dark. Cells were viewed using a laser-scanning confocal microscope (Leica SP8, Weztlar, CA, USA). For morphometric analysis, the mitochondrial fragmentation counts (MFC) were analyzed using ImageJ 1.8.0 software (Bethesda, Maryland, MD, USA). The Manders’ overlap coefficient was used to measure the co-localization between mitochondria and COX-2, p-Drp1^Ser616^, or SIRT3 using IPP 6.0 (Media Cybernetics, Silver Spring, MD, USA).

### 2.11. Proximity Ligation Assay (PLA)

PLA was performed by Duolink^®^ In Situ Detection Reagents (Sigma Aldrich), as previously described [13]. Briefly, after being treated, cells mounted on coverslides were fixed with 4% PFA and permeabilized with 0.5% Triton X-100. The coverslides were blocked in Duolink II solution for 1 h and incubated with the primary antibodies anti-COX-2 (1:200) and anti-p-Drp1^Ser616^ (1:400) at 4 °C overnight, followed by incubation of Duolink PLA anti-Mouse PLUS or anti-Rabbit PLUS proximity probes. Before counterstaining with mounting medium containing 4′,6-diamidino-2-phenylindole (DAPI) (1:200), the samples were incubated with the ligase (1:40) at 37 °C for 30 min and then polymerase (1:80) for 100 min. PLA signals’ foci formation of COX-2 and p-Drp1^Ser616^ was visualized using a laser-scanning confocal microscope (Leica SP8).

### 2.12. I-TASSER Analysis

Iterative threading assembly refinement (I-TASSER) is a hierarchical protocol for automated protein structure prediction and structure-based function annotation (platform URL: https://zhanglab.ccmb.med.umich.edu/I-TASSER/, access on 5 November 2019). The three-dimensional (3D) structures of COX-2 and Drp1, and the potential structure of the COX-2-Drp1 interacted complex, were predicted using the I-TASSER server and visualized using the open-source molecular graphics program PyMOL [33].

### 2.13. Cell Viability, EdU Incorporation Assay, and Apoptosis Analysis

Cell viability was measured using MTS assay with the kit (Promega, Madison, WI, USA), as described previously [32]. Briefly, after being treated, the microplates were read using a microplate spectrophotometer system (BMG LabTech, Ortenberg, Germany) at a wavelength of 490 nm. Cell viability rate (%) = (OD_490Sample_ − OD_490Blank_)/(OD_490Ctrl_ − OD_490Blank_) × 100%.

For EdU incorporation assay, cells were stained for EdU (20 μM) using the BeyoClick™ EdU cell proliferation kit (Beyotime, Shanghai, China). Cells mounted on coverslides were fixed with 4% PFA and permeabilized with 0.3% Triton X-100. Subsequently, click reaction solution was prepared following the protocol described and incubated with coverslides for 30 min. Hoechst 33342 was used for nuclear staining. The positive rate of EdU labeling cells was analyzed using ImageJ software (Bethesda, Maryland, MD, USA).

An Annexin V-FITC/PI Apoptosis Detection Kit (Beyotime, Shanghai, China) was used for apoptosis detection, as in our previous study [32]. After being treated, cells were harvested and resuspended in Annexin V binding buffer and stained with Annexin V-FITC and propidium iodide (PI). Subsequently, stained cells were analyzed using flow cytometry (Beckman Coulter, Indianapolis, IN, USA). The data were analyzed using FlowJo v.7.6.5 (Tree Star Inc., Ashland, OR, USA).

### 2.14. Statistical Analyses

Statistical analyses were performed with the Statistical Package for Social Sciences (SPSS) version 16.0 (IBM Corp, Armonk, NY, USA). All grouped data were presented as means ± standard deviation (SD) from at least three independent experiments. The statistical significance was determined using a two-tailed unpaired Student’s *t* test (*t*-test) for comparison between two groups. Multiple groups’ comparison was determined by one-way analysis of variance (ANOVA) followed by Dunnett’s *t*-test. A value of * *p* < 0.05, ** *p* < 0.01, or *** *p* < 0.001 was considered a significant difference.

## 3. Results

### 3.1. Upregulation of COX-2 and Drp1 Is Associated with the Poorer Prognosis of HCC Patients

To clarify the relationship between COX-2 and Drp1 in HCC patients, we collected and analyzed 12 paired HCC tumor (T) tissues and peritumor (P) tissues. Hematoxylin and eosin (HE) staining revealed cancer nests in tumor tissues (Figure 1A and Appendix A). Serial section analysis of immunohistochemical (IHC) staining showed that the expression levels of COX-2 and Drp1 were upregulated in HCC tumor (T) tissues compared to the corresponding peritumor (P) tissues (Figure 1B and Appendix A). Moreover, the co-expression of COX-2 and Drp1 was visually confirmed by immunofluorescence (IF). We found that the intensity of COX-2 and Drp1 expression was upregulated, and the co-localization of COX-2 and Drp1 was increased in HCC tumor (T) tissues compared to the corresponding peritumor (P) tissues (Figure 1C). Western blot analysis also confirmed the upregulation of COX-2 and Drp1 relative expression in HCC tumor tissues, while the quantitation and trend relationship analyses indicated that the levels of COX-2 and Drp1 were upregulated in tumor tissues compared to peritumor tissues, and there was a positive correlation between the levels of COX-2 and Drp1 expression in HCC sample tissues (*r* = 0.7558, *p* < 0.0001) (Figure 1D,E).

To investigate the variations in gene expression associated with HCC progression, an RNA-seq dataset (accession no.: GSE104310; Appendix A) from HCC samples (*n* = 20) was used for differentially expressed genes’ (DEGs’) identification. A total of 13,548 genes was identified as either up- or downregulated. By choosing cutoffs at ≥1.5 or ≤−1.5 log2 (fold change) and *p* < 0.05, we found 150 genes upregulated (red plots) and 358 genes downregulated (green plots) in HCC, relative to normal tissues (Figure 1F). Hierarchical cluster analysis (HCA) of mitochondria-related DEGs showed that the relative levels of expression of *PTGS2* (encoding COX-2) and *DNM1L* (encoding Drp1) genes were increased in HCC samples compared to normal tissues (Figure 1G). Based on the screened DEGs, including the upregulated *PTGS2* gene in HCC, bioinformatics analyses were used to explore the role of *PTGS2*-related signaling pathways’ regulation in hepatocarcinogenesis. Gene ontology (GO) enrichment analysis identified pathways significantly affected by the differential expression of *PTGS2* in the hepatocarcinogenesis process. As a result, we found that the pathways related to the “response to drug”, “signal transduction”, and “protein binding” were among the enriched terms in COX-2 high-expression-associated HCC tumor tissues (Appendix A). Kyoto Encyclopedia of Genes and Genomes (KEGG) pathway analysis of DEGs was conducted according to the gene ratio, number of genes, and *p* value. The results further demonstrated that COX-2-related pathways, including “chemical carcinogenesis”, “drug metabolism-cytochrome P450”, “metabolism of xenobiotics by cytochrome P450”, and “arachidonic acid metabolism” sets, were significantly enriched in HCC tumor tissues (Appendix A). Moreover, we performed single-gene set enrichment analysis (GSEA) to explore the significant signaling pathways regulated by the expression of *PTGS2* and *DNM1L* genes between low- and high-expression groups. We mainly focused on the mitochondria-related signals, especially the mitochondrial fission pathway. It was found that the normalized enrichment scores (NESs) for the mitochondrial fission signals were 1.58 and 1.63 (*p* < 0.05), respectively, which suggested that *PTGS2* and *DNM1L* expression was positively correlated with mitochondrial fission (Figure 1H). Furthermore, to test if *PTGS2* expression was correlated to that of *DNM1L*, we analyzed one additional RNA-seq dataset (accession no.: GSE36376; Appendix A). The analysis revealed that the relative expressions of *PTGS2* and *DNM1L* were remarkably upregulated in HCC tumor tissues (*n* = 240) compared with normal samples (*n* = 193) (Appendix A). GSEA showed that the NESs for the mitochondrial fission signals were 1.11 and 1.20, respectively, which suggested that the expression levels of the *PTGS2* and *DNM1L* genes were positively correlated with mitochondrial fission (Appendix A). Moreover, the results from GEO datasets revealed a positive correlation between the expression of *PTGS2* and *DNM1L* genes in these clinical HCC samples (*r* = 0.31, *p* = 0.038 for GSE104310; *r* = 0.22, *p* < 0.001 for GSE36376, respectively) (Appendix A). These bioinformatics results indicated that the regulation of COX-2 expression might be involved in promoting hepatocarcinogenesis via the relationship with Drp1-dependent mitochondrial fission. Importantly, Dong et al. showed that high COX-2 levels led to worse overall survival of HCC patients [34]. Here, the expressions of COX-2 and Drp1 were also negatively associated with overall survival using the Kaplan-Meier plotter (*p* = 0.041 and *p* = 0.017, respectively) (Figure 1I). Taken together, these data indicated that the correlation of COX-2 and Drp1 upregulation might contribute to mitochondrial fission and was associated with a poorer prognosis in HCC patients.

### 3.2. Activation of Drp1 Enhances Mitochondrial Fission and Its Molecular Association with COX-2 in HCC Cells

To explore the potential role of COX-2 in Drp1-mediated mitochondrial fission of HCC cells, we firstly identified the mRNA and protein expression of Drp1 and COX-2 in four HCC cell lines (QGY7703, HepG2, Huh7, and MHCC97H) and hepatic cell lines (L02 and QSG7701) by qRT-PCR and western blotting. The expression of COX-2 and Drp1 was remarkably high in HCC cell lines compared to hepatic cell lines (Appendix A). L02, HepG2, and MHCC97H cells were selected to establish different mitochondrial fission cell models. We characterized the mitochondrial dynamics in cells by transmission electron microscopy (TEM). TEM images revealed that the mitochondrial length was relatively shorter in two HCC cell lines (HepG2 and MHCC97H) compared to L02 cells, suggesting that mitochondrial fission was increased in HCC cells (Figure 2A and Appendix A). IF imaging confirmed that the mitochondrial fission rate was higher in HCC cells than in L02 cells (Figure 2B and Appendix A). The p-Drp1^Ser616^ is an activating event that contributes to the OMM localization of Drp1 and subsequent mitochondrial fission, while the p-Drp1^Ser637^ is a deactivation modification [14]. The results showed that the endogenous expression levels of COX-2, Drp1, and p-Drp1^Ser616^ were remarkably higher, while p-Drp1^Ser637^ was lower, in HCC cell lines (HepG2 and MHCC97H) than in L02 cells (Figure 2C). Therefore, we hypothesized that COX-2 might be a driver of p-Drp1^Ser616^ activation to promote hepatocarcinogenesis.

To elucidate the necessity of p-Drp1^Ser616^ for the relationship between COX-2 and Drp1, the novel kinases-dependent mechanism for p-Drp1^Ser616^ activation underlying mediation of the mito-COX-2/p-Drp1^Ser616^ interaction was investigated. Multipattern kinases, which were known to potentially regulate p-Drp1^Ser616^, including PKCα, p38 MAPK, CDK2, ERK, AKT, and the mitochondrial kinase PINK1, were screened. We found that the levels of the abovementioned kinases were relatively increased in HepG2 and MHCC97H cells compared to L02 cells (Figure 2D). The result supported the notion that Drp1 activation through Ser616 phosphorylation was regulated by a multi-kinase framework. To further investigate the role of a special mitochondrial kinase on regulating p-Drp1^Ser616^, we focused on mitochondrial localization of PINK1 to elucidate the mechanism implicating mitochondrial protein machinery including mito-COX-2 and p-Drp1^Ser616^ in the mitochondrial dynamics of HCC cells. Interestingly, our results suggested that the endogenous level of mitochondrial kinase PINK1 was significantly increased in HCC cells (Figure 2D). Furthermore, serial section analysis of IF showed that the expression levels of PINK1 and p-Drp1^Ser616^ were upregulated, and the colocalization of PINK1 and p-Drp1^Ser616^ was increased in HCC tumor (T) tissues compared to the corresponding peritumor (P) tissues (Appendix A). Bioinformatics analysis also showed that *PINK1* was positively correlated with *PTGS2*, or *DNM1L* in HCC samples from GSE49515 (*r* = 0.572 or 0.374, respectively, *p* < 0.05) (Appendix A). Our results suggested that PINK1 might be involved in the activation of p-Drp1^Ser616^ with the induction of COX-2. Next, we isolated mitochondrial fractions from L02, HepG2, and MHCC97H cells to study the subcellular localization of related mitochondrial protein machinery. It was shown that localization of COX-2, PINK1, and p-Drp1^Ser616^ in endogenous mitochondrial fractions was relatively higher in proportion, and p-Drp1^Ser637^ was lower in proportion, in HepG2 and MHCC97H cells than in L02 cells (Figure 2E). Protein-protein interaction (PPI) network analysis was performed to identify mitochondrial dynamics-related proteins likely to interact with COX-2. Analyses indicated a close relationship between the *PTGS2* and *DNM1L* genes (Appendix A), which revealed a potential direct interaction of COX-2 and Drp1. Furthermore, the primary sequence and the 3D structures of Drp1, COX-2, and the COX-2/Drp1 complex were generated using the I-TASSER server and PyMOL for molecular docking analysis (Figure 2F and Appendix A). As shown in Figure 2G, the potential binding areas involved in COX-2/Drp1 interaction were predicted, including the amino acid sequence 80–125 aa and 71–114 aa, 197–229 aa and 272–315 aa, 292–317 aa and 404–428 aa, and 702–724 aa and 483–504 aa. These results supported the existence of a direct interaction between COX-2 and Drp1. Furthermore, endogenous mitochondrial protein immunoprecipitation (mito-IP) assay showed that the interaction of mito-COX-2, PINK1, and p-Drp1^Ser616^ was increased in HCC cells. Moreover, the results showed that the increased phosphorylation of Drp1 at Ser616 was regulated by PINK1 in the mitochondria, which interacted with mito-COX-2, thereby promoting p-Drp1^Ser616^-driven mitochondrial fission (Figure 2H).

To further verify that the phosphorylation of Drp1 at site Ser616 is necessary for the COX-2/Drp1 interaction, and that the increased phosphorylation is PINK1-dependent, we constructed a gain-of-function of Drp1 S616A mutation (alanine site-directed mutation from serine, S to A, to block phosphorylation) using recombinant expression plasmids in HepG2 cells (named HepG2-*DNM1L*(S>A) cells). The results showed that carbonyl cyanide-m-chlorophenyl-hydrazine (CCCP, a PINK1 agonist) treatment induced an increase of p-Drp1^Ser616^ in HepG2-*DNM1L* cells compared to HepG2-*DNM1L*(S>A) cells (Figure 2I). Furthermore, we found that the interaction of mitochondrial COX-2, PINK1, and p-Drp1^Ser616^ was decreased in HepG2-*DNM1L*(S>A) cells (Figure 2J). Taken together, these results provided p-Drp1^Ser616^-based evidence linking a novel PINK1-mediated p-Drp1^Ser616^ activation mechanism for the regulation of mito-COX-2 to the PINK1/p-Drp1^Ser616^ interaction and its co-localization in endogenous mitochondrial protein machinery, which further modulated the mitochondrial fission in HCC cells.

### 3.3. Mito-COX-2 Modulates Mitochondrial Fission by Stabilizing the Activity of p-Drp1^Ser616^ in HCC Cells

We established stable overexpression of COX-2 by transfection of HepG2 and MHCC97H cells with a *PTGS2* expression vector, to confirm the role of COX-2 in mitochondrial fission. COX-2 overexpression in HepG2-pB-*PTGS2* and MHCC97H-pB-*PTGS2* cells was conducted (Figure 3A). Overexpression of COX-2 led to increasing clonogenicity, as evaluated by cell colony formation assay (Figure 3B). Immunofluorescence revealed that COX-2-overexpression increased the level of p-Drp1^Ser616^, mitochondrial fragmentation counts, and mito-COX-2 and p-Drp1^Ser616^ co-localization in HCC cells (Figure 3C). Next, we examined the subcellular localization of COX-2, PINK1, Drp1, and p-Drp1^Ser616^ in COX-2 overexpressing HCC cells, and observed that overexpression of COX-2 markedly increased the redistribution of COX-2, PINK1, and p-Drp1^Ser616^ to mitochondria (Figure 3D). Furthermore, in situ proximity ligation assay (PLA) indicated that the gain-of-function of COX-2 in HCC cells increased the formation and number of red fluorescence foci, indicative of enhanced COX-2/p-Drp1^Ser616^ proximity, and thus increased the interaction between these two proteins in the mitochondria (Figure 3E).

The requirement of mito-COX-2 for p-Drp1^Ser616^-dependent remodeling of mitochondrial dynamics in HCC cells was further studied. As depicted in Figure 3F, we constructed a plasmid driving the expression of COX-2 protein, containing a mitochondrial targeting sequence (MTS) tagged to its N-terminal and a Flag tagged to its C-terminal (pBabe-MTS-*PTGS2*-Flag). This construction ensured the retention of COX-2 by the mitochondria, allowing mitochondrial protein immunoprecipitation (mito-IP). After transfecting pBabe-MTS-*PTGS2*-Flag to HepG2 and MHCC97H cells, mito-IP was performed using an anti-Flag antibody, and the precipitate was analyzed by western blotting with antibodies against PINK1 and p-Drp1^Ser616^. As demonstrated in Figure 3G, the levels of PINK1 and p-Drp1^ser616^ were increased in the cells overexpressing mito-COX-2. To confirm the role of mito-COX-2-modulated mitochondrial fission in the survival outcome of HCC cells, cell proliferation was measured with the EdU incorporation assay. The results showed that COX-2-overexpressing HCC cells had significantly more EdU incorporation than those in control cells (Figure 3H,I). We further explored the effect of mito-COX-2 overexpression on mitochondria-dependent apoptosis, and cell viability was measured. The results showed that cell viability increased, and the pro-apoptotic proteins Bax and cleaved Caspase-3 decreased, while the anti-apoptotic protein Bcl-2 increased in MTS-driven mito-COX-2-overexpressing HCC cells (Appendix A). Taken together, these results implied an increased activity of p-Drp1^Ser616^ phosphorylated by PINK1 to promote the interaction of mito-COX-2/p-Drp1^Ser616^. This mechanism could explain the enhanced mitochondrial fission-modulated MQC, which regulated cell proliferation in mito-COX-2-overexpressed HCC cells.

### 3.4. Suppression of Mito-COX-2 Translocation Decreases Its Interaction with p-Drp1^Ser616^ and Modulates Mitochondrial Fission in HCC Cells

To investigate whether reduced COX-2 expression would decrease mito-COX-2/p-Drp1^Ser616^ interaction, and consequently, suppress Drp1-mediated mitochondrial fission, we generated stable COX-2-knockdown HepG2- and MHCC97H-Cas9-*PTGS2* cell lines by CRISPR/Cas9-based gene editing (Figure 4A,B). According to the knockout efficiency, the cell model constructed by the sgRNA-2 recombinant plasmid was used in the experiment. The expression levels of PINK1 and p-Drp1^Ser616^ in the total proteins of whole-cell lysates were significantly decreased, while COX-2 was suppressed in COX-2-knockdown cells (Figure 4C). As shown in Figure 4D, COX-2-knockdown in HCC cells resulted in a marked decrease of mito-COX-2 and p-Drp1^Ser616^ translocation to mitochondria, and mitochondrial fragmentation counts. We also found that the proportions of mitochondrial COX-2, PINK1, and p-Drp1^Ser616^ were markedly downregulated in COX-2-knockdown cells (Figure 4E). The proximity of COX-2 and p-Drp1^Ser616^ was reduced in COX-2-knockdown cells, as visualized by the PLA foci formation and number (Figure 4F). Next, we isolated mitochondrial fractions from COX-2-knockdown cells to study the interaction of COX-2 with PINK1 and p-Drp1^Ser616^. Mito-IP was performed using the anti-COX-2 antibody, and results showed that the PINK1-activated mito-COX-2/p-Drp1^Ser616^ interaction in the mitochondria was reduced in COX-2-knockdown of HepG2- and MHCC97H-Cas9-*PTGS2* cells (Figure 4G). These results, obtained in COX-2-knockdown cells, indicated that inhibiting mito-COX-2 translocation decreased the distribution of Drp1 and the activity of p-Drp1^Ser616^, to promote mitochondrial fission. Knockdown of COX-2 led to reducing self-renewal and proliferation, as evaluated by cell colony formation assay (Appendix A). Furthermore, the results showed that COX-2-knockdown HCC cells had significantly less EdU incorporation than control cells (Appendix A). We further explored the effect of COX-2 knockdown on mitochondrial-dependent apoptosis. The results showed that cell viability decreased, and the pro-apoptotic proteins Bax and cleaved Caspase-3 increased, while the anti-apoptotic protein Bcl-2 decreased in COX-2-knockdown HCC cells (Appendix A). Taken together, we cross-validated the data obtained in COX-2-overexpression HCC cells (Figure 3) and CRISPR/Cas9-based COX-2-knockdown HCC cells (Figure 4), and these results highlighted that the regulatory mechanism of PINK1-activated p-Drp1^Ser616^ mediated the functional stability of mito-COX-2/p-Drp1^Ser616^ interaction and its dependent mitochondrial fission, which was linked to modulating the survival phenotypes and outcomes of HCC cells in vitro (Figure 4H).

### 3.5. Suppression of HCC Xenograft Growth via the Inhibition of p-Drp1^Ser616^ by Targeted Intervention on Mito-COX-2 Translocation In Vivo

The effect of COX-2-coupled Drp1-driven mitochondrial fission on HCC cell growth was studied in vivo using a xenograft nude mouse model. As shown in Figure 5A, mice implanted with MTS-directed mito-COX-2-overexpressing HepG2-MTS-*PTGS2*-Flag cells exhibited faster tumor growth, a larger size, and higher tumor weight than control animals. In contrast, mice implanted with COX-2-knockdown HepG2-Cas9-*PTGS2* cells showed slower tumor growth, a smaller size, and lower tumor weight than mice implanted with HepG2-Cas9-NC cells (Figure 5B). Moreover, immunofluorescence analysis of tissue sections showed that HCC xenografts overexpressing mito-COX-2 had significantly higher percentages of COX-2- and p-Drp1^Ser616^-positive cells and an increased co-localization of COX-2 and p-Drp1^Ser616^ than those made of HepG2-pB-MTS-Flag cells (Figure 5C). In contrast, HCC xenografts with COX-2-knockdown exhibited a significantly lower percentage of COX-2- and p-Drp1^Ser616^-positive cells and reduced co-localization of COX-2 and p-Drp1^Ser616^ than those made of HepG2-Cas9-NC cells (Figure 5D). Furthermore, we examined the subcellular localization of COX-2 and p-Drp1^Ser616^ in HCC cells harboring xenografts. It was found that the distribution and proportion of COX-2, PINK1, and p-Drp1^Ser616^ in mitochondrial fractions were increased in COX-2-overexpressing HepG2 xenografts and decreased in COX-2-knockdown HepG2 xenografts (Figure 5E,F). These results suggested that mito-COX-2-dependent mitochondrial fission accelerated HCC growth via the regulatory mechanism underlying the PINK1-activated p-Drp1^Ser616^ mitochondrial translocation and mito-COX-2/p-Drp1^Ser616^ interaction, and targeting the intervention to mito-COX-2 could inhibit p-Drp1^Ser616^-driven HCC growth in vivo.

### 3.6. Targeted Intervention on Mito-COX-2 Enhances Chemosensitivity by Inhibiting p-Drp1^Ser616^-Driven Mitochondrial Fission in Platinum Drug-Treated HCC Cells

Previous evidence indicates that increased mitochondrial fission alleviates the processes of cell death and decreases the chemotherapeutic efficacy [18,19]. We tested whether mito-COX-2 would affect the chemosensitivity by regulating p-Drp1^Ser616^-driven mitochondrial fission in HCC cells. Three commonly used platinum chemotherapy drugs—cDDP, CBP, and L-OHP—were administered to HCC cells. Concentration- and time-dependent decreases in cell viability were measured by MTS assay (Appendix A). Dose-dependent accumulation of COX-2, Drp1, and p-Drp1^Ser616^ was also observed in HCC cells treated with platinum drugs (Figure 6A). As visualized by IF imaging, treatment with platinum drugs induced the levels of COX-2 expression and increased COX-2 translocation to the mitochondria (increase of COX-2-mitochondria colocalized staining), while the mitochondrial fission was shown with the increase of mitochondrial fragmentation counts (Figure 6B,C). In MTS-directed mito-COX-2-overexpressing HCC cells, the expression of mito-COX-2, PINK1 and p-Drp1^Ser616^ was further increased on treatment with platinum drugs (Figure 6D). Flow cytometric analysis indicated that on cDDP treatment, the apoptosis rates were about 17.7% ± 0.4% and 13.0% ± 0.3%, respectively, in HepG2- and MHCC97H-pBabe control cells. However, overexpression of COX-2 decreased the apoptosis rates to 9.4% ± 0.3% and 6.3% ± 0.4% in cDDP-treated HepG2- and MHCC97H-MTS-*PTGS2*-Flag cells, respectively (Figure 6E). These results suggested that platinum drugs induced the level of COX-2 expression and increased localization of mito-COX-2, which might be involved in the inhibition of apoptosis by inducing PINK1 to mediate the activation of p-Drp1^Ser616^. Therefore, we further adopted the Cas9-*PTGS2*-based COX-2-knockdown HCC cell model to induce apoptosis-sensitive outcomes. In COX-2-knockdown HCC cells, the expression of mito-COX-2, PINK1, and p-Drp1^Ser616^ was decreased on treatment with platinum drugs (Figure 6F). The apoptosis rates were about 10.6% ± 0.6% and 13.1% ± 0.5% in cDDP-treated HepG2- and MHCC97H-Cas9 control cells, whereas they increased to 24.5% ± 0.2% and 24.1% ± 0.2%, respectively, in cDDP-treated COX-2-knockdown HepG2- and MHCC97H-Cas9-*PTGS2* cells (Figure 6G). Collectively, these findings indicated that in HCC cells treated with platinum drugs, targeted inhibition of mito-COX-2 enhanced the chemosensitivity by suppressing the mito-COX-2/p-Drp1^Ser616^ interaction mediating mitochondrial fission.

### 3.7. Suppression of Drp1 Promotes Apoptosis via Inhibition of Mito-COX-2/p-Drp1^Ser616^ Interaction in Platinum Drug-Treated HCC Cells

In this study, we focused on addressing the impact of novel functional regulation of mito-COX-2/p-Drp1^Ser616^ interaction in the mitochondria on hepatocarcinogenesis and platinum drug-induced anti-cancer against HCC. The regulatory effects of targeting Drp1 suppression, such as the selective knockdown (KD) of Drp1 (*DNM1L* gene) by the pharmaceutical inhibitor (Mdivi-1) and genetic intervention (si*DNM1L*), were determined. In platinum chemotherapy drug (cDDP, CBP, and L-OHP)-administered HepG2 cells, the expression of COX-2 was induced, and the levels of Drp1 and p-Drp1^Ser616^ increased (Figure 7A). Moreover, it was shown that Mdivi-1 significantly reduced the increased levels of Drp1 and p-Drp1^Ser616^, but had no effect on the COX-2 level (Figure 7A). Co-localization mitochondria/p-Drp1^Ser616^ and mitochondrial fission in platinum drug-treated HepG2 cells were inhibited by Drp1 intervention with Mdivi-1 (Figure 7B, Right panel), compared with cells without Mdivi-1 treatment (Figure 7B, Left panel). As shown by the increased Manders’ overlap coefficient of mitochondria/p-Drp1^Ser616^ and mitochondrial fragmentation counts, platinum drugs markedly increased the mitochondrial fission and associated p-Drp1^Ser616^ translocation from the cytoplasm to the mitochondria in HepG2 cells, while these decreased in Mdivi-1 treatment groups (Figure 7B,C). Increased PLA foci and red fluorescence intensities showed that COX-2/p-Drp1^Ser616^ proximity was enhanced in cDDP-treated HepG2 cells, compared to control cells. However, the increased fluorescence intensity and the formation of PLA foci were reduced on Mdivi-1 administration (Figure 7D). The subcellular localization of p-Drp1^Ser616^ and COX-2 was examined in both mitochondrial and cytoplasmic fractions isolated from HepG2 cells, whether treated or untreated with cDDP. Treatment with cDDP increased the p-Drp1^Ser616^ and COX-2 mitochondrial distribution, while such effect was reduced by Mdivi-1 pretreatment (Figure 7E). Furthermore, mito-IP assay showed that cDDP-induced mito-COX-2/p-Drp1^Ser616^ interaction was reduced in cells pretreated with Mdivi-1, compared with cells treated with cDDP alone (Figure 7E). To study the roles of the inhibition of mito-COX-2/p-Drp1^Ser616^ interaction-driven mitochondrial fission by Mdivi-1, mitochondria-dependent apoptosis was monitored to evaluate the sensitivity of HepG2 cells to chemotherapy. Pretreatment with Mdivi-1 markedly increased the apoptosis in cells treated with platinum drugs (cDDP, CBP, and L-OHP), compared to cells treated with platinum drugs alone (Appendix A). On Mdivi-1 intervention, platinum drug-induced apoptosis was promoted with the increase of the pro-apoptotic proteins Bax and cleaved Caspase-3 and the reduction of anti-apoptotic protein Bcl-2 (Figure 7F).

The functional effect of Drp1 inhibition was further assessed by using small interfering RNA (siRNA) to specifically suppress *DNM1L* transcripts. It was shown that si*DNM1L* significantly reduced the increased levels of COX-2, Drp1, and p-Drp1^Ser616^ induced by platinum drugs’ (cDDP, CBP, and L-OHP) administration in HepG2 cells (Appendix A). p-Drp1^Ser616^ translocation from the cytosol to the mitochondria was markedly reduced in si*DNM1L* cells treated with platinum drugs, compared to cells without si*DNM1L* intervention (Appendix A). In situ PLA showed that si*DNM1L* significantly reduced the cDDP-mediated increase of COX-2/p-Drp1^Ser616^ proximity (Figure 7G). Furthermore, the subcellular localization of p-Drp1^Ser616^ and COX-2 was examined in both cytoplasmic and mitochondrial fractions. It was shown that treatment with cDDP increased the proportion of p-Drp1^Ser616^ and the COX-2 mitochondrial distribution, while pretreatment with si*DNM1L* reversed this effect (Figure 7H). Moreover, mito-IP assay showed that cDDP-induced mito-COX-2/p-Drp1^Ser616^ interaction was reduced in cells pretreated with si*DNM1L*, compared to cells receiving cDDP alone (Figure 7H). On si*DNM1L* intervention, platinum drug-induced apoptosis was promoted with the increase of the pro-apoptotic proteins Bax and cleaved Caspase-3 and the reduction of anti-apoptotic protein Bcl-2 (Figure 7I). It was shown that the results obtained by using si*DNM1L*, corresponding to Drp1 gene-targeted intervention, were in good agreement with those obtained from experiments using the Drp1 inhibitor Mdivi-1. Taken together, these results indicated that targeting the intervention to Drp1 would be involved in suppressing the mito-COX-2/p-Drp1^Ser616^ interaction and its driven mitochondrial fission, which potentiates the pro-apoptotic effect and anti-tumor sensitivity in HCC cells treated with platinum drugs (Appendix A).

### 3.8. Deacetylation of Mito-COX-2 via SIRT3 Activation Mediates Higher Sensitivity of HCC to Cisplatin by Inhibiting Mito-COX-2/p-Drp1^Ser616^ Interactions In Vitro and In Vivo

Accumulating evidence indicates that mitochondrial protein acetylation is a widespread and key regulatory mechanism for the PTMs of mitochondrial proteins [35]. Mitochondrial protein acetylation is controlled mostly by the nicotinamide adenine dinucleotide (NAD)^+^-dependent mitochondrial protein deacetylase sirtuin 3 (SIRT3). Studies have suggested that SIRT3 activation enhances the therapeutic efficacy of chemotherapy [28]. To explore whether mito-COX-2/p-Drp1^Ser616^ interaction was regulated by COX-2 deacetylation, and whether targeting mito-COX-2 deacetylation would sensitize HCC cells to cDDP, RSV was used for in vitro and in vivo studies. RSV, one of the natural phytochemicals, has proven beneficial for mitochondrial function partly via SIRT3, through increasing mitochondrial protein deacetylation [36,37]. In the current study, we investigated whether RSV sensitizes HCC cells to cDDP, and whether SIRT3-mediated mito-COX-2 deacetylation is involved in this process. Cell viability was decreased in a dose-dependent manner in HCC cells treated with RSV (Appendix A). Co-treatment of HepG2 cells and MHCC97H cells with RSV and cDDP resulted in increased expression of SIRT3 and reduced expression of COX-2 and p-Drp1^Ser616^, compared to cDDP alone (Appendix A). RSV countered the effect of cDDP in increasing mitochondrial translocation of COX-2 and mitochondrial fission in both HepG2 and MHCC97H cells, and the co-localization of mito-COX-2/p-Drp1^Ser616^ was decreased in cells treated with RSV and cDDP (Figure 8A and Appendix A). Furthermore, mito-IP assay using the anti-SIRT3 antibody showed that mito-COX-2/p-Drp1^Ser616^ interaction was reduced in RSV-treated HCC cells and correlated with increased SIRT3 (Figure 8B). Moreover, mito-IP assay using the anti-COX-2 antibody showed that the level of acetylated mito-COX-2 was decreased, which correlated with reduced mito-COX-2/p-Drp1^Ser616^ interaction in RSV-treated cells (Figure 8C). These results suggested that the stability of the mito-COX-2/p-Drp1^Ser616^ complex was regulated through SIRT3-mediated mito-COX-2 deacetylation. Co-treatment of RSV and cDDP for 24 h further decreased the cell viability, suggesting that SIRT3-mediated mito-COX-2 deacetylation was involved in the therapeutic activity of platinum drugs (Appendix A). On RSV administration, the cDDP-induced apoptosis was promoted with the increased levels of Bax and cleaved Caspase-3 and the reduction of Bcl-2 (Figure 8D), while flow cytometry analysis showed that co-treatment with RSV and cDDP significantly increased the proportion of apoptotic cells, compared to cDDP treatment alone (Appendix A). These results indicated that RSV promoted SIRT3-mediated mito-COX-2 deacetylation, and this mechanism may be involved in the inducible apoptotic cell death of HCC cells. The results were confirmed in si*SIRT3*-treated HpG2 cells. Furthermore, in constructed SIRT3-overexpressing HCC cells, the cDDP-induced levels of pro-apoptotic proteins Bax and cleaved Caspase-3 were significantly increased and the reduction of anti-apoptotic protein Bcl-2 was decreased compared to control cells and cells treated with cDDP alone (Appendix A). Consistently, flow cytometry analysis showed that the proportion of apoptotic cells was significantly increased in cDDP-treated SIRT3-overexpressing HCC cells compared to cDDP-treatment alone (Appendix A). These findings proved that SIRT3 activation reduced cDDP-induced acetylation of COX-2, which in turn, suppressed mito-COX-2/ p-Drp1^Ser616^ interaction, and subsequently sensitized HCC cells to the mitochondria-dependent apoptosis.

To further demonstrate the role of SIRT3 in the modification of COX-2 deacetylation in vivo, we tested the effect of RSV on mito-COX-2 deacetylation and the resulting anti-tumor sensitivity to cDDP in the HepG2 cell xenograft-harboring nude mice model. Combined RSV and cDDP administration led to a significant reduction in tumor volume, size, and weight compared to control mice or mice treated with RSV or cDDP alone (Figure 8E and Appendix A). Western blot analysis showed an induction of SIRT3 expression and reduction of COX-2 and p-Drp1^Ser616^ mitochondrial distribution in xenograft tumors with combined RSV and cDDP therapy compared to cDDP alone (Figure 8F and Appendix A). IF assay showed that SIRT3 was induced, and a reduction of COX-2 and SIRT3/COX-2 co-localization was suppressed in xenograft tumors from the RSV administration, suggesting SIRT3 was involved in the regulation of COX-2 deacetylation in cDDP-treated HCC xenograft tumors (Appendix A). It was shown that COX-2/p-Drp1^Ser616^ co-localization was reduced in xenograft tumors from the combined RSV and cDDP therapy group, compared to cDDP alone (Figure 8G). Furthermore, it was found that co-treatment with RSV and cDDP upregulated the levels of Bax and cleaved Caspase-3, and downregulated the level of Bcl-2, compared to cDDP alone (Figure 8H). These results obtained in vivo were in agreement with the findings in vitro, suggesting that deacetylation of mito-COX-2 exerted selective PTM toward mitochondrial protein quality control (MPQC), which was a potential target to achieve HCC cell-growth inhibition. Cumulatively, our results suggested that SIRT3 activation could regulate the acetylation status and stability of mito-COX-2 translocated to mitochondria, mediating the mito-COX-2/p-Drp1^Ser616^ interaction to promote the mitochondrial fission-driven functional outcomes and anti-tumor sensitivity of HCC cells in vitro and in vivo. Regulation of the SIRT3-mito-COX-2-p-Drp1^Ser616^ signaling axis is a novel mechanism underlying the theranostics for targeting interventions in hepatocarcinogenesis.

## 4. Discussion

In the current study, we presented evidence that COX-2 was frequently overexpressed in HCC, and more importantly, that the PINK1-mediated mito-COX-2/p-Drp1^Ser616^ interaction played a multifaceted role by promoting mitochondrial fission-driven functional outcomes and anti-tumor sensitivity of HCC cells in vitro and in vivo. One of the key findings of our study was that mito-COX-2 deacetylation, triggered by SIRT3, regulated the stability of mito-COX-2 translocation to mitochondria and its interaction with p-Drp1^Ser616^. Targeting the suppression of mito-COX-2 and selective induction of SIRT3 could inhibit p-Drp1^Ser616^-driven mitochondrial fission, to induce apoptosis and sensitize HCC cells to multipattern anti-tumor therapy. Our study indicated that mito-COX-2/p-Drp1^Ser616^ interaction was the regulatory hub for the anti-cancer balance between the processes of COX-2-dependent cell survival and apoptotic cell death in HCC. Thus, mito-COX-2 homeostasis may represent a valuable theranostic biomarker and treatment target for HCC, and possibly for other types of COX-2-high-expression cancers.

Over the past decades, COX-2 appeared to be a double-edged sword, acting as a pro- or anti-cancer enzyme. Oncogenic properties of COX-2 have been found in several types of cancer, including breast cancer, colorectal cancer, and HCC [38,39,40,41,42]. For instance, COX-2 overexpression causes apoptotic resistance in colorectal cancer cells (HCT15 and HT29) by activating Hippo-Yes-associated protein (YAP) and p38/COX-2/PGE2 pathways and inducing Bcl-2 expression [39,40]. Zha et al. [42] reported that melatonin sensitizes HepG2 cells to ER stress-induced apoptosis by inhibiting COX-2 expression, increasing the levels of CHOP, and decreasing the Bcl-2/Bax ratio. In this report, we found that COX-2 and Drp1 were frequently upregulated in HCC tumor tissues. Distinct from prior research, we also found that the proportions of mitochondrial COX-2 and Drp1 were increased in HCC tissues and cell lines, and indicated that the relative expression level of COX-2 was positively correlated with that of Drp1, while Drp1 expression was also negatively associated with overall survival of HCC patients. In a steady state, mitochondrial networks in cells result from a balance between mitochondrial fission and fusion as the dynamics of MQC [43,44]. The phosphorylation of Drp1 at Ser616 (p-Drp1^Ser616^) is an activating event contributing to the translocation of Drp1 from the cytoplasm to the OMM, which in turn, promotes mitochondrial fission and mitophagy. This process acts as a surveillance system for the phosphorylated forms of p-Drp1^Ser616^ and MQC [45]. Mitochondrial protein quality control (MPQC), a key process regulating MQC, plays a critical role in maintaining the component and distribution of mitochondrial protein machinery in the OMM and compartment [46]. Several mechanisms have been proposed to regulate MPQC, including PTMs of mitochondrial proteins. Dysregulated PTMs of mitochondrial proteins alter the normal functions of these processes such as cell growth and apoptosis, which can contribute to the development of disease, including HCC [47]. Chung et al. [48] revealed that Drp1 activation through Ser616 phosphorylation was regulated by ERK/AKT and CDK2 in lung adenocarcinoma cell lines. It has been shown that the multi-kinase framework including ERK/AKT and CDK2 promotes cell proliferation of A549 and HCC827 cells by activating Drp1 [48]. Tang et al. [49] found that COX-2 contributed to the upregulation of mitochondrial transcription factor A (TFAM), mediated by p38-MAPK activation, and the subsequent enhancement of Drp1-dependent mitochondrial fission in human osteosarcoma U-2 OS and cervical cancer HeLa cells. However, the mechanisms remain to be elucidated for how p-Drp1^Ser616^ is regulated in HCC cells with high COX-2 expression. In this report, our results showed that the levels of COX-2, Drp1, and p-Drp1^Ser616^ were remarkably high in the mitochondria of HCC cell lines, indicating that COX-2 might be a driver of Drp1 activation to promote hepatocarcinogenesis. Furthermore, our results showed that increased p-Drp1^Ser616^ was regulated by PINK1 in the mitochondria, which interacted with mito-COX-2, thereby increasing p-Drp1^Ser616^-mediated mitochondrial fission and cell proliferation in HCC.

Recently, the subcellular localization of COX-2 was suspected to promote or suppress cancers’ progression and resistance to apoptosis. The activation of extracellular-regulated kinases 1 and 2 (ERK1/2) induces the nuclear accumulation (shuttling) of COX-2 [50]. The nuclear shuttling of the ERK/COX-2 complexes may facilitate p53 phosphorylation at Ser15 and subsequent apoptosis in ovarian cancer cells [51]. In other words, cytoplasmic retention of COX-2 leads to ovarian cancer cell proliferation and is mediated by inhibition of ERK/COX-2 complex formation. Our previous studies have demonstrated that COX-2 is present in mitochondria (mito-COX-2) and that suppression of COX-2 translocation to mitochondria inhibits mitochondrial fission, cancer stemness, and apoptosis resistance in nasopharyngeal carcinoma cells [24]. Altogether, these studies showed that subcellular accumulation in different components, such as the differential retention of COX-2 in the cytoplasm or mitochondria, acts as a promoter of cancer progression. In the present study, we identified that PINK1-mediated p-Drp1^Ser616^ regulates the interaction of mito-COX-2 with activated Drp1, which further enhances mitochondrial fission in HCC cells. Importantly, inducible mito-COX-2/p-Drp1^Ser616^ interaction increases in mitochondria. Thus, mito-COX-2 may be a new chaperone protein for p-Drp1^Ser616^ in mitochondria to modulate the survival phenotypes of HCC cells. Furthermore, we for the first time established the oncogenic role of mito-COX-2 in HCC in vitro and in vivo. Mechanistic analysis demonstrated that a significant increase of mito-COX-2 translocation and mito-COX2/p-Drp1^Ser616^ interaction correlates with increased mitochondrial fission, cell proliferation, and resistance to apoptosis in HCC.

Although elevated levels of COX-2 have been widely related to inflammation-related carcinogenesis, high variability exists among individuals [52]. Less is known about the PTMs of COX-2, which can regulate its function and subcellular localization. Our previous studies have shown that p-COX-2^Ser601^ undergoes ER retention for proteasome degradation. Moreover, dephosphorylation of COX-2 by protein phosphatase 2A (PP2A)-B55δ promotes its translocation from the ER to mitochondria, and subsequent inflammatory response in hepatocytes [26]. Growing evidence suggests that MPQC-mediated mitochondrial dynamics play a central and multifunctional role in malignant tumor progression [53,54,55]. Mitochondrial protein acetylation is widespread and a key PTM regulatory mechanism of MPQC. SIRT3 resides in the mitochondria and represents a multifunctional deacetylase regulating the acetylation state of the mitochondrial protein involved in MQC [56]. Liao et al. [57] found that honokiol treatment increased the ability of SIRT3 to repress the expression of COX-2 and pro-inflammatory cytokines, thus mitigating radiation-induced brain injury in a zebrafish model. SIRT3 deficiency elevated the expression of IL-1β, TNF-α, and COX-2, and increased microglial proliferation and activation [58]. Therefore, further elucidation of the mechanisms underlying SIRT3-regulated mito-COX-2 acetylation for its MPQC is of great interest. In the present study, RSV was used to induce the upregulation of SIRT3 endogenous expression in HCC cells and xenograft tumors. Combined with genetic (SIRT3-overexpression) and pharmacological (RSV) interventions, our results were in keeping with data showing that SIRT3 activation reduced the acetylation status and stability of mito-COX-2 translocation to mitochondria. Interestingly, as a consequence, SIRT3-dependent mito-COX-2 deacetylation prevented its interaction with p-Drp1^Ser616^, followed by the sensitivity of mitochondria-dependent apoptosis in HCC. Here, for the first time, our data demonstrated that regulation of the SIRT3/mito-COX-2/p-Drp1^Ser616^ signaling axis was the novel mechanism underlying the targeted intervention in hepatocarcinogenesis.

“Mitochondrial Medicine” recognizes the important role played by mitochondria in cancer. Targeting MQC for cancer therapy requires the targeting of mitochondrial protein machinery with specific functions, including the MPQC [59,60]. The current work, which investigated the mechanisms of mito-COX-2 regulation, demonstrated the central role played by PTMs with deacetylation. We used phytochemical agent (RSV) and genetic (SIRT3 over-expression) interventions to mimic the induction of SIRT3 activation and investigate the MPQC regulation of mito-COX-2 distribution and function. In the current study, we established that SIRT3-mediated deacetylation of mito-COX-2 inhibited its mitochondrial localization and decreased mito-COX-2/p-Drp1^Ser616^ stability, and it repressed p-Drp1^Ser616^-driven mitochondrial fission. Our results suggested that the increased SIRT3 expression repressed the level of mito-COX-2 through PTMs of its deacetylation, mediating the inhibition of mito-COX-2 interaction with p-Drp1^Ser616^ and potentiating the chemosensitivity of HCC cells to platinum drugs in vitro and in vivo. To explore the common mechanism by which MQC intervention affects anti-HCC sensitivity, platinum drugs (cDDP, CBP, L-OHP) were used to establish chemotherapy models. In the current study, COX-2-knockdown based on CRISPR/Cas9 suppressed mito-COX-2/p-Drp1^Ser616^ interaction and increased the chemosensitivity of HCC. Consistent with our results, Tang et al. found that inhibition of COX-2 by NS-398 sensitized cancer cells to radiotherapy via the downregulation of the p38/Drp1/TFAM signaling axis, mediating mitochondrial fission [49]. Suppression of Drp1-driven mitochondrial fission enhanced the sensitivity of hypoxic ovarian cancer cells to cDDP [61]. In this study, we focused on investigating the impact of novel functional regulation of mito-COX-2/p-Drp1^Ser616^ interaction in the mitochondria on hepatocarcinogenesis and how to target anti-cancer interventions against HCC. Our results showed that Drp1 was concomitantly upregulated with COX-2 during platinum drug treatment in HCC cells. Using si*DNM1L* and the Drp1 inhibitor, we provided evidence that p-Drp1^Ser616^-mediated mitochondrial fission offered an explanation for cell survival and chemoresistance in platinum drug-treated HCC cells. These data indicated that targeting the suppression of SIRT3/mito-COX-2/p-Drp1^Ser616^ signaling axis-associated MQC regulation succeeds in addressing a novel, multi-pattern, anti-tumor target with which to increase chemosensitivity in HCC.

## 5. Conclusions

In summary, we demonstrated that the high-expression of COX-2 and Drp1 contributed to MQC regulation associated with a poorer prognosis in HCC. PINK1 mediated the activation of p-Drp1^Ser616^ and modulated the interaction of mito-COX2/p-Drp1^Ser616^ and its driven mitochondrial fission. SIRT3 was a regulator for the deacetylation of mito-COX-2, which was involved in the MPQC of mito-COX2/p-Drp1^Ser616^-related mitochondrial protein machinery. Targeting the SIRT3/mito-COX-2/p-Drp1^Ser616^ signaling axis promoted MQC-dependent chemosensitivity via multi-pattern, anti-tumor mechanisms in HCC. Our findings established both a biological and clinical rationale supporting the use of mito-COX-2 as a theranostic biomarker and potential target to modulate cellular survival or apoptosis in hepatocarcinogenesis.

## Figures and Tables

**Figure 1 cancers-14-00821-f001:**
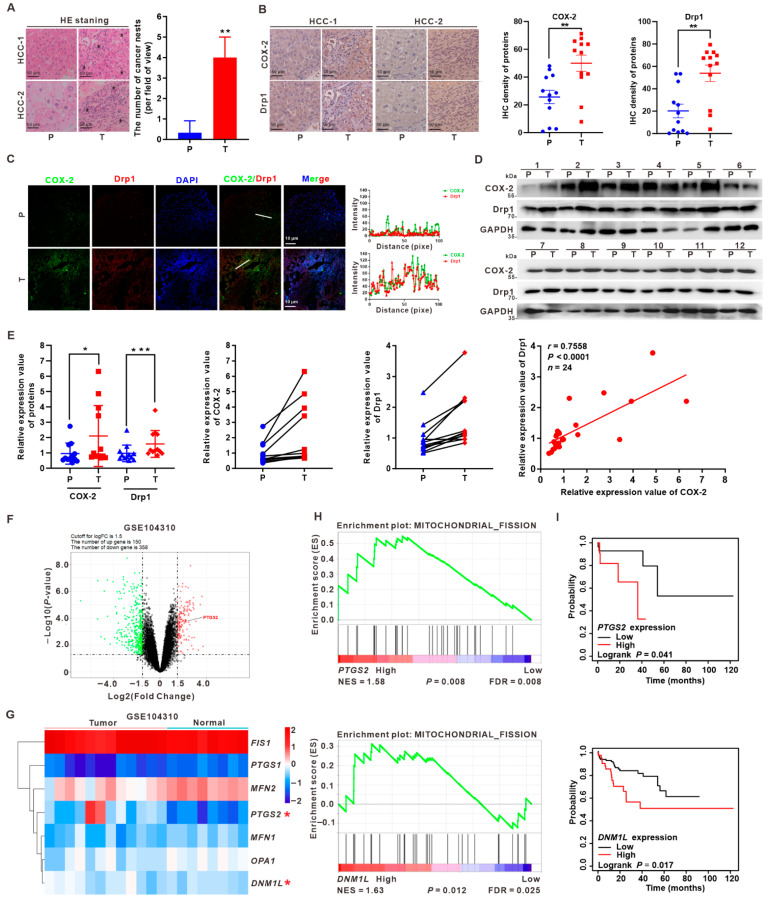
Upregulation of COX-2 and Drp1 is associated with a poorer prognosis of HCC patients. (**A**–**E**) Twelve paired tumor and peritumor tissues from HCC patients were collected and analyzed. (**A**) Representative hematoxylins and eosin (HE) staining images of the paired peritumor (P) and tumor (T) tissues from the HCC samples of #1 and #2 (Left). Black asterisks indicate cancer nests. Scale bar, 50 μm. The number of cancer nests per field of view is quantified in the bar graph (Right). Data are shown as the mean ± SD, *n* = 12. ** *p* < 0.01 as compared to the peritumor (P) tissue group. (**B**) Representative IHC staining images for COX-2 and Drp1 expression in the paired peritumor (P) and tumor (T) tissues from the HCC samples of #1 and #2 (Left). Scale bar, 50 μm. The scatter plot graph for the quantitative analysis of COX-2 and Drp1 expression by IPP 6.0 is shown (Right). Data are shown as the mean ± SD, *n* = 12. ** *p* < 0.01 as compared to the peritumor (P) tissue group. (**C**) Serial sections of the paired peritumor (P) and tumor (T) tissues were subjected to immunofluorescence analysis to evaluate the co-expression of COX-2 (green) and Drp1 (red), and the co-localization (yellow) between COX-2 and Drp1 (Left). DAPI (Blue) was for nucleus staining. Fluorescence curves were generated using Zen 2010 software (Right). Scale bars, 10 μm. (**D**) Western blotting analysis for the protein levels of COX-2 and Drp1 expression in 12 pairs of peritumor (P) and tumor (T) tissues. Full Western Blot images can be found in Appendix A. (**E**) The scatter plot graph for the quantitative analysis of COX-2 and Drp1 expression (Left) and the trend relationship analysis of COX-2 and Drp1 expression (Middle), in the pairs of peritumor (P) and tumor (T) tissues. Correlation between the expression of COX-2 and Drp1 was calculated by Pearson’s correlation analysis (Right). Data are shown as the mean ± SD, *n* = 12. * *p* < 0.05, *** *p* < 0.001 as compared to the peritumor (P) tissue group. (**F**–**H**) Gene expression analysis of RNA-seq dataset (accession no.: GSE104310) in tumor and normal tissues samples from HCC patients (*n* = 20). (**F**) Volcano plot of 13,548 genes in total. The DEGs with the fold change ≥1.5 or ≤−1.5, *p* < 0.05 are presented in the red plots (upregulated genes, *n* = 150) and green plots (downregulated genes, *n* = 358). Black plots represent the rest of the genes (*n* = 13,040) with no significant expression change. The upregulated *PTGS2* gene is shown. (**G**) Heatmap for the relative expression levels of mitochondria-related DEGs. The significant increases of *PTGS2* and *DNM1L* in HCC tumor samples are shown. * *p* < 0.05, compared to the normal group. (**H**) GSEA indicates significant correlations between *PTGS2* and *DNM1L* genes’ expression and mitochondrial fission signatures. (**I**) Kaplan-Meier curve analysis of overall survival in HCC patients by the expression of *PTGS2* and *DNM1L* using the Kaplan-Meier plotter.

**Figure 2 cancers-14-00821-f002:**
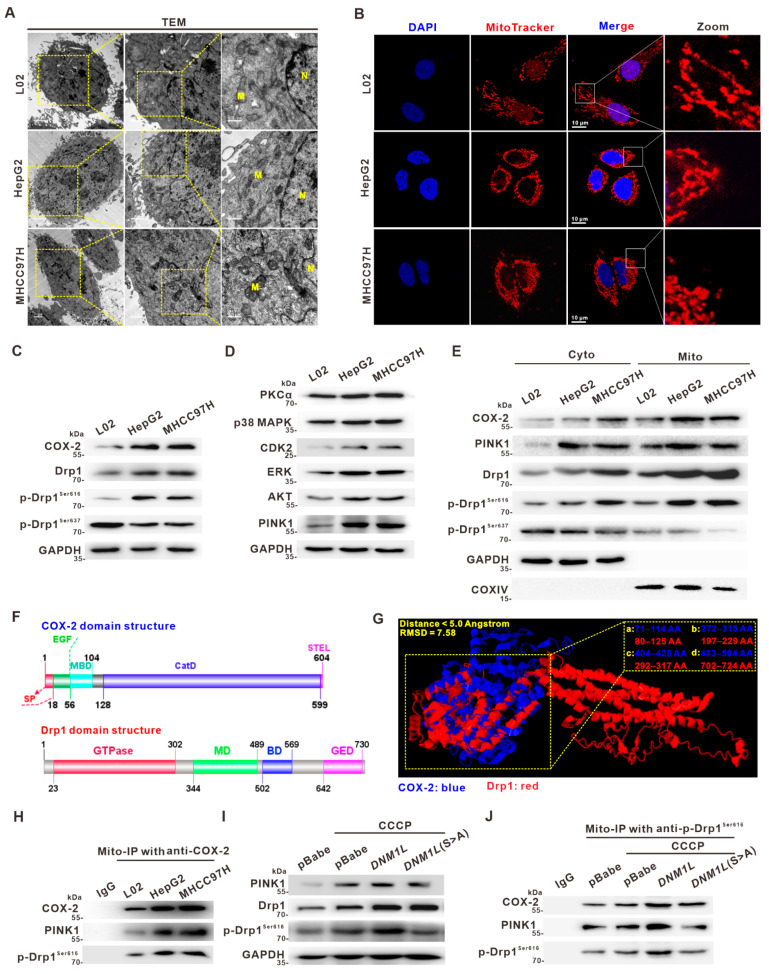
Activation of Drp1 enhances mitochondrial fission and its interaction with COX-2 in HCC cells. (**A**) Transmission electron microscopy (TEM) photomicrographs of mitochondrial structure in L02, HepG2, and MHCC97H cells. Scale bar, 2 μm (Left), 1 μm (Middle), and 500 nm (Right). (**B**) Mitochondrial morphology was analyzed by confocal microscopy. (**C**–**E**) The expression levels of indicated proteins in L02, HepG2, and MHCC97H cells were measured by western blot. (**C**) The levels of COX-2, Drp1, p-Drp1^Ser616^, and p-Drp1^Ser637^. (**D**) The levels of PKCα, p38 MAPK, CDK2, ERK, AKT, and PINK1. (**E**) Cytoplasmic (Cyto) and mitochondrial (Mito) fractions were prepared and analyzed the levels of COX-2, PINK1, Drp1, p-Drp1^Ser616^, and p-Drp1^Ser637^. (**F**) Schematic illustration of the primary sequence of human COX-2 and Drp1 and their predicted domains in molecular structures are displayed. Featured COX-2 conserved domains: SP, signal peptide; EGF, EGF domain; MBD, membrane-binding domain; CatD, catalytic domain; STEL, an inefficient ER retention signal. Featured Drp1 conserved domains: GTPase, G domain; MD, middle domain; VD, variable domain; GED, GTPase effector domain. (**G**) Protein-protein structures of the COX-2/Drp1 complex were predicted using I-TASSER. Potential binding areas are indicated by dashed boxes. (**H**) Mitochondrial fractions were immunoprecipitated with anti-COX-2 antibody, and the expressions of PINK1 and p-Drp1^Ser616^ were detected by western blot. IgG was used as the control. (**I**,**J**) HepG2-pBabe, HepG2-*DNM1L*, and HepG2-*DNM1L*(S>A) cells were constructed and treated with or without 20 μM CCCP for 4 h. (**I**) The levels of PINK1, Drp1, and p-Drp1^Ser616^ were detected by western blot. (**J**) Mitochondrial fractions were immunoprecipitated with anti-p-Drp1^Ser616^ antibody, and the levels of p-Drp1^Ser616^, COX-2, and PINK1 were detected by western blot. IgG was used as the control. Full Western Blot images can be found in Appendix A.

**Figure 3 cancers-14-00821-f003:**
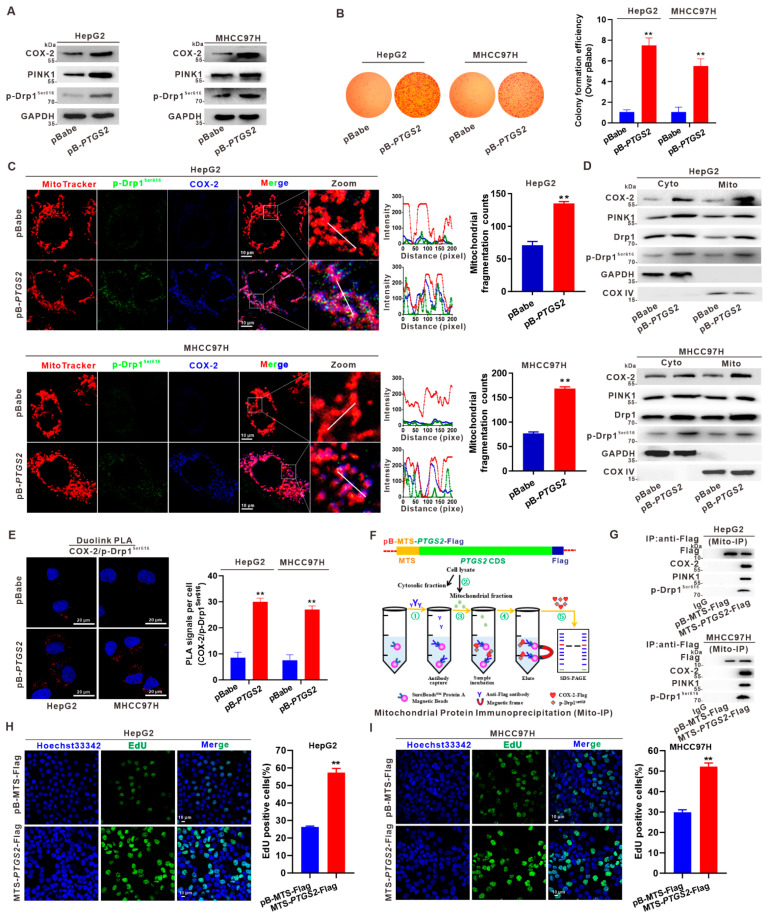
Mito-COX-2 modulates mitochondrial fission by stabilizing the activity of p-Drp1^Ser616^ in HCC cells. (**A**–**E**) HepG2- and MHCC97H-pB-*PTGS2* cells with COX-2-overexpression were established, together with HepG2- and MHCC97H-pBabe control cells. (**A**) Expression levels of COX-2, PINK1, and p-Drp1^Ser616^ proteins in cells were detected by western blot. (**B**) The self-renewal capacity of cells was measured by colony formation assay. Representative colony formation images (Left) and the efficiency of colony formation (Right) are shown. (**C**) Immunofluorescence images showing the co-localization of mitochondria (red), p-Drp1^Ser616^ (green), and COX-2 (blue) were captured by confocal microscopy (Left). Scale bars, 10 μm. Fluorescence curves were generated using Zen 2010 software (Middle). The mitochondrial fragmentation counts calculated by IPP 6.0 are shown in the bar graphs (Right). (**D**) Cytoplasmic (Cyto) and mitochondrial (Mito) fractions were prepared and subjected to western blot. (**E**) Representative PLA images show the interaction between COX-2 and p-Drp1^Ser616^ (Left). Scale bar: 20 μm. Quantification of COX-2/p-Drp1^Ser616^ PLA signals per cell is shown in the bar graph (Right). Data are expressed as the mean ± SD. ** *p* < 0.01, compared to the corresponding pBabe control cells. (**F**–**I**) HepG2- and MHCC97H-MTS-*PTGS2*-Flag cells overexpressing mito-COX-2 were established, together with HepG2- and MHCC97H-pB-MTS-Flag control cells. (**F**) Construction of a pBabe-MTS-*PTGS2*-Flag plasmid driving the overexpression of mitochondria-targeted resident COX-2 (Upper). The mitochondrial targeting sequence (MTS) tagged to the N-terminal and a Flag tagged to the C-terminal of *PTGS2*. Schematic diagram detailing the mito-IP assay (Lower). (**G**) Mitochondrial fractions were immunoprecipitated (mito-IP) with an anti-Flag antibody, and the levels of the indicated proteins were detected to display the interaction between mito-COX-2, PINK1, and p-Drp1^Ser616^ in cells. IgG was used as the control. (**H**,**I**) Cell proliferation was evaluated by EdU incorporation assay (Left). Scale bar, 10 μm. Quantification of EdU-positive cells’ ratio is shown in the bar graph (Right). Data are expressed as the mean ± SD. ** *p* < 0.01, compared to the corresponding control cells. Full Western Blot images can be found in Appendix A.

**Figure 4 cancers-14-00821-f004:**
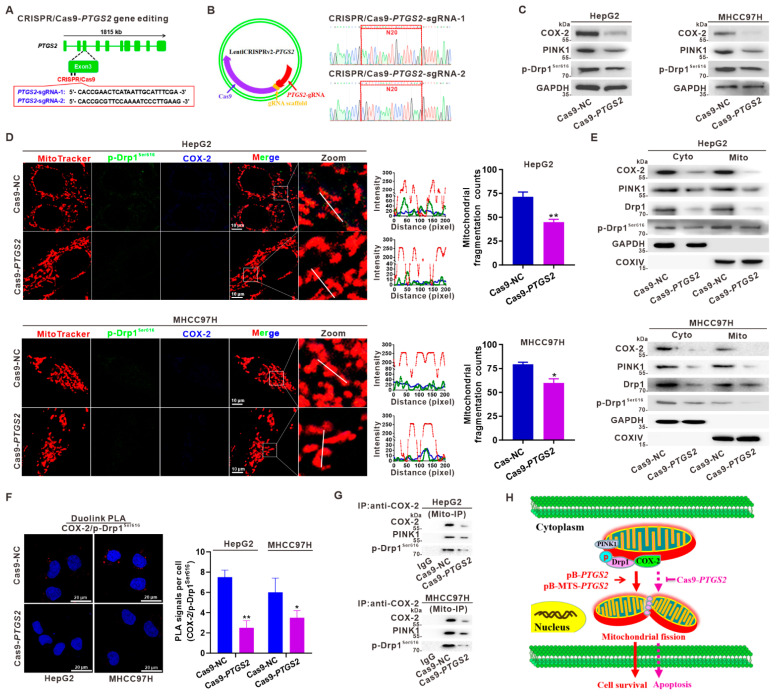
Suppression of mito-COX-2 translocation decreases its interaction with p-Drp1^Ser616^ and modulates mitochondrial fission in HCC cells. (**A**) Schematic diagram of *PTGS2*-gRNAs’ design for CRISPR/Cas9-based gene editing targeted to *PTGS2*. (**B**) Construction of COX-2-knockdown lentiCRISPRv2-*PTGS2* recombinant plasmid (Left), and confirmation by sequencing (Right). (**C**–**G**) HepG2- and MHCC97H-Cas9-*PTGS2* cells with COX-2-knockdown were established, together with HepG2- and MHCC97H-Cas9-NC control cells. (**C**) Expression levels of COX-2, PINK1, and p-Drp1^Ser616^ in total proteins of whole-cell lysates were detected by western blot. (**D**) Immunofluorescence images showing the co-localization of mitochondria (red), p-Drp1^Ser616^ (green), and COX-2 (blue) were captured by confocal microscopy (Left). Scale bars, 10 μm. Fluorescence curves were generated using Zen 2010 software (Middle). The mitochondrial fragmentation counts calculated by IPP 6.0 are shown in the bar graphs (Right). (**E**) Cytoplasmic (Cyto) and mitochondrial (Mito) fractions were prepared and subjected to western blot. (**F**) Representative PLA images showing the interaction between COX-2 and p-Drp1^Ser616^ (Left). Scale bar, 20 μm. Quantification of the COX-2/p-Drp1^Ser616^ PLA signals per cell is shown in the bar graph (Right). Data are expressed as the mean ± SD. * *p* < 0.05, ** *p* < 0.01, compared to Cas9-NC control cells. (**G**) Mitochondrial fractions were immunoprecipitated (mito-IP) with anti-COX-2 antibody, and the levels of proteins were detected to indicate the protein interaction between mito-COX-2, PINK1, and p-Drp1^Ser616^ in COX-2-knockdown HepG2- and MHCC97H-Cas9-*PTGS2* cells. IgG was used as the control. (**H**) Schematic representation showing the impact of regulation of mito-COX-2 on the functional stability of p-Drp1^Ser616^-dependent mitochondrial fission in HCC cells. Full Western Blot images can be found in Appendix A.

**Figure 5 cancers-14-00821-f005:**
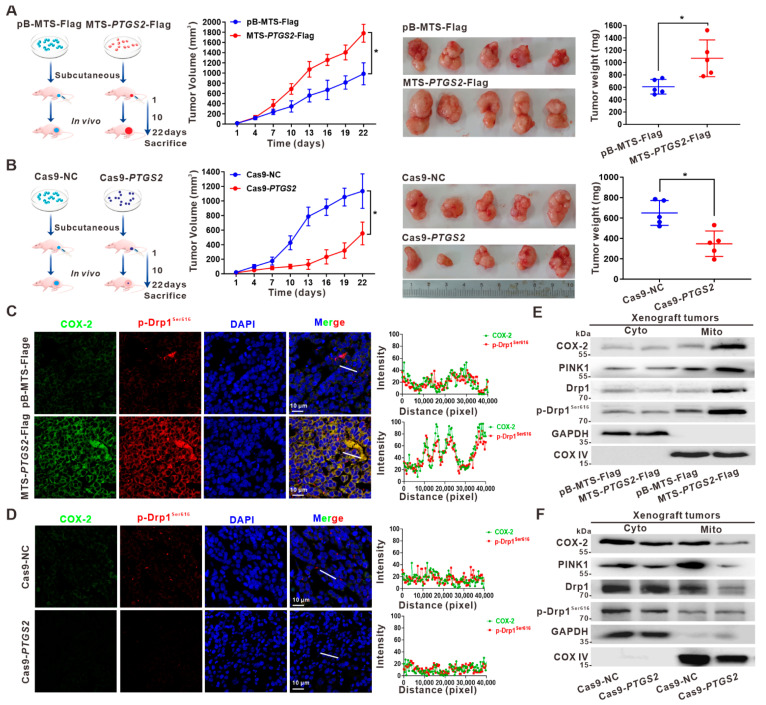
Suppression of HCC xenograft growth via the inhibition of p-Drp1^Ser616^ by targeted intervention on mito-COX-2 translocation in vivo. Tumorigenicity was analyzed in BALB/c nude mice subcutaneously injected with HepG2-MTS-*PTGS2*-Flag or HepG2-Cas9-*PTGS2* cells (1 × 10^6^/mouse). (**A**,**B**) Xenograft nude mouse model (Left), growth curves of the xenograft tumors’ volume (Middle-Left), representative images of the xenograft tumors’ size (Middle-Right), and tumor weights (Right) are shown. *n* = 5/group. * *p* < 0.05, compared to the corresponding HepG2-pB-MTS-Flag and HepG2-Cas9-NC group. (**C**,**D**) Serial sections of xenograft tumors were subjected to immunofluorescence analysis to evaluate the expression of COX-2 (green), p-Drp1^Ser616^ (red), and the co-localization (yellow) between COX-2 and p-Drp1^Ser616^ (Left). DAPI (Blue) was for nucleus staining. Fluorescence curves were generated using Zen 2010 software (Right). Scale bars, 10 μm. (**E**,**F**) Cytoplasmic (Cyto) and mitochondrial (Mito) fractions were prepared from xenografts where the tumors harbored mito-COX-2-overexpressing HepG2 cells (**E**) and the COX-2-knowndown HepG2 cells (**F**), and were subjected to western blot. Full Western Blot images can be found in Appendix A.

**Figure 6 cancers-14-00821-f006:**
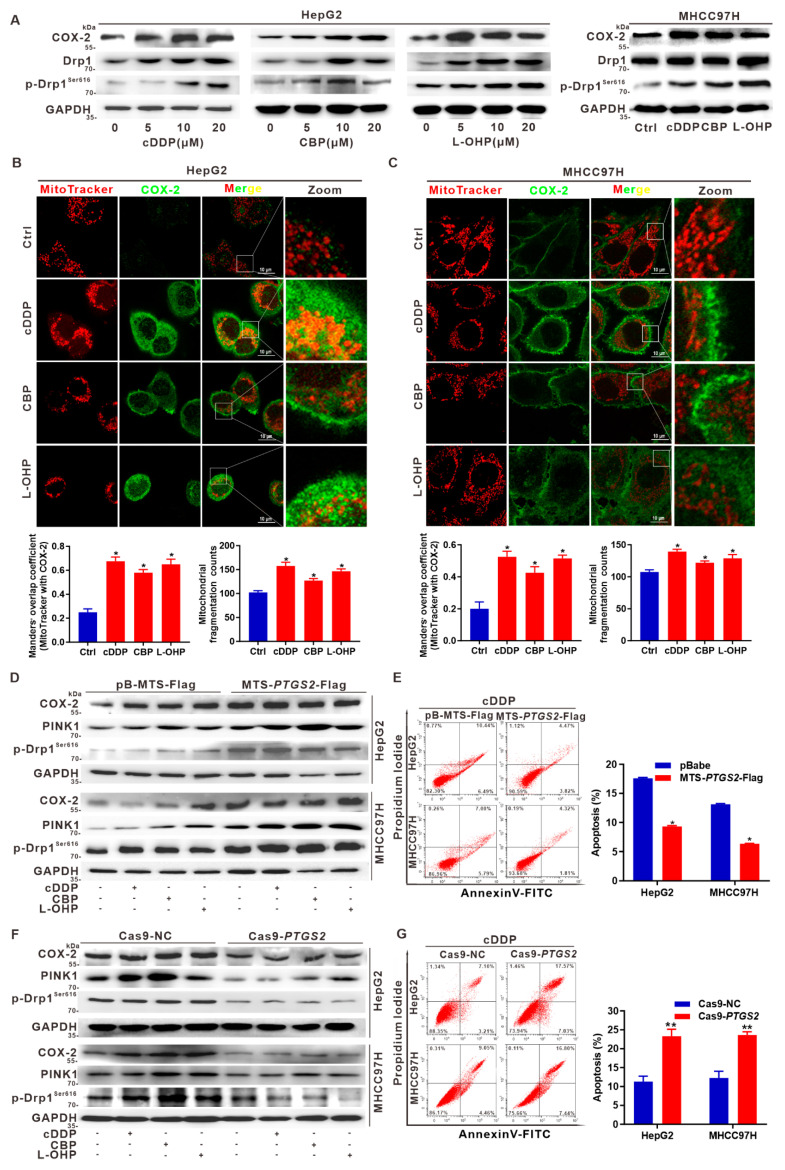
Targeted intervention on mito-COX-2 enhances chemosensitivity by inhibiting p-Drp1^Ser616^-driven mitochondrial fission in platinum drug-treated HCC cells. Three commonly used platinum chemotherapy drugs—cDDP, CBP, and L-OHP—were administered in HCC cells. (**A**) HepG2 and MHCC97H cells were treated with the indicated concentrations of platinum drugs for 12 h. The expression of COX-2, Drp1, and p-Drp1^Ser616^ was evaluated by western blot. HepG2 (**B**) and MHCC97H (**C**) cells were treated with platinum drugs (10 μM) for 12 h. Representative IF images of MitoTracker (red), COX-2 (green), and their co-localization (yellow) were produced using confocal microscopy (Upper). Scale bars, 10 μm. Manders’ overlap coefficients for co-localization of COX-2 with mitochondria and mitochondrial fragmentation counts were calculated using IPP 6.0, while their quantification is shown in the bar graphs (Lower). * *p* < 0.05, compared to the corresponding control groups. (**D**,**E**) MTS-directed mito-COX-2 overexpressing HepG2- and MHCC97H-MTS-*PTGS2*-Flag cells were treated with or without platinum drugs (10 μM) for 12 h. (**D**) Whole-cell lysates were subjected to western blotting to measure the levels of COX-2, PINK1, Drp1, and p-Drp1^Ser616^. (**E**) Cells were treated with or without cDDP (10 μM) for 12 h. Cells were stained with Annexin V-FITC/PI and the apoptosis rate was detected by flow cytometry. The representative analysis of apoptosis by flow cytometry is shown (Left). The quantification of apoptotic cells is shown in the bar graph (Right). * *p* < 0.05, compared to the corresponding control groups. (**F**,**G**) CRISPR/Cas9-based COX-2-knockdown HepG2- and MHCC97H-Cas9-*PTGS2* cells were treated with or without platinum drugs (10 μM) for 12 h. (**F**) Whole-cell lysates were subjected to western blotting to measure the levels of COX-2, PINK1, Drp1, and p-Drp1^Ser616^. (**G**) Cells were treated with or without cDDP (10 μM) for 12 h. Cells were stained with Annexin V-FITC/PI and the apoptosis rate was detected by flow cytometry. The representative analysis of apoptosis by flow cytometry is shown (Left). The quantification of apoptotic cells is shown in the bar graph (Right). Data are expressed as the mean ± SD. ** *p* < 0.01, compared to the corresponding control groups. Full Western Blot images can be found in Appendix A.

**Figure 7 cancers-14-00821-f007:**
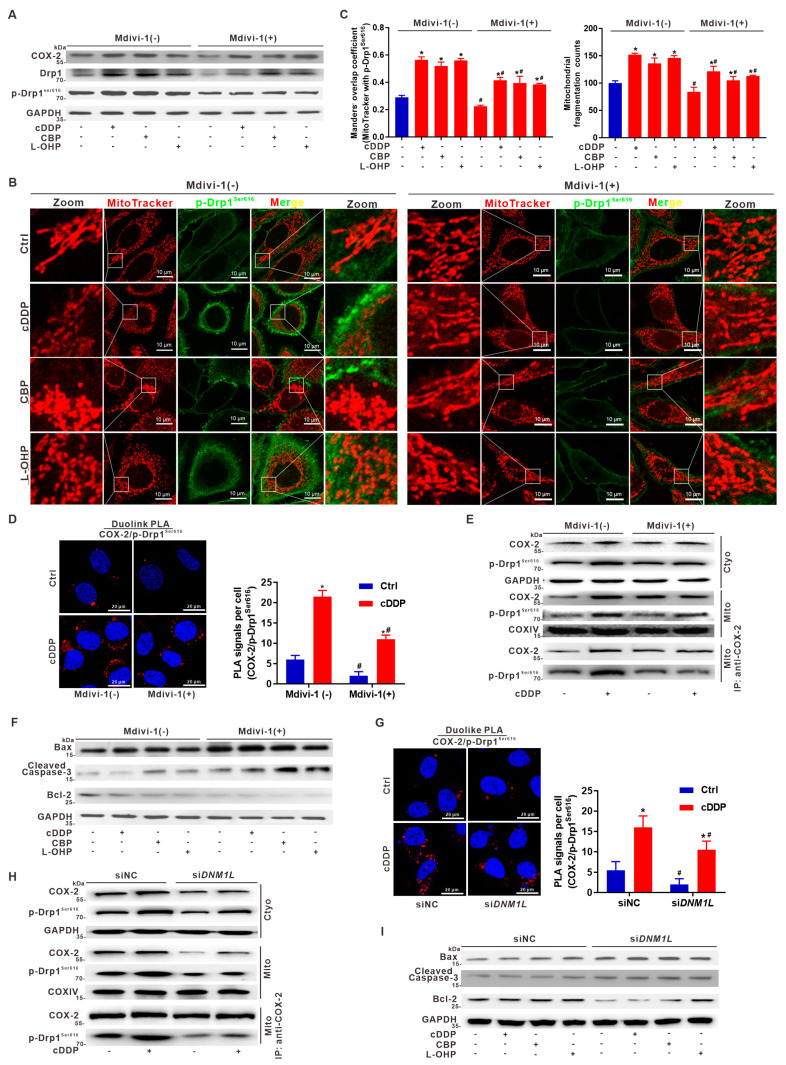
Suppression of Drp1 promotes apoptosis via inhibition of mito-COX-2/p-Drp1^Ser616^ interaction in platinum drug-treated HCC cells. Three commonly used platinum chemotherapy drugs—cDDP, CBP, and L-OHP—were administered to HCC cells. (**A**–**F**) HepG2 cells were pre-treated with or without Mdivi-1 (20 μM) for 12 h, followed by platinum drug treatment for 12 h. (**A**) Whole-cell lysates were subjected to western blotting to measure the indicated proteins. (**B**,**C**) Subcellular localization of p-Drp1^Ser616^ in cells was examined using a confocal microscope. (**B**) Representative images of mitochondria (red), p-Drp1^Ser616^ (green), and their co-localization (yellow) are shown. Scale bars, 10 μm. (**C**) Manders’ overlap coefficients for co-localization of p-Drp1^Ser616^ with mitochondria (Left) and mitochondrial fragmentation counts (Right) were calculated using IPP 6.0, while the quantification is shown in the bar graph. (**D**) Representative PLA images showing the interaction between COX-2 and p-Drp1^Ser616^ (Left). Scale bar, 20 μm. Quantification of COX-2/p-Drp1^Ser616^ PLA signals per cell is shown in the bar graph (Right). Data are expressed as the mean ± SD. * *p* < 0.05, compared to the corresponding control groups. ^#^ *p* < 0.05, compared to the corresponding Mdivi1(-) groups. (**E**) Cytoplasmic (Cyto) and mitochondrial (Mito) fractions were prepared and subjected to western blot analysis (Upper and Middle). Mitochondrial fractions were immunoprecipitated with anti-COX-2 antibody (Lower). The level and proportion of COX-2 and p-Drp1^Ser616^ were detected. (**F**) The levels of apoptosis-related proteins Bax, cleaved Caspase-3, and Bcl-2 were determined by western blotting. (**G**–**I**) HepG2 cells were pre-treated with or without si*DNM1L* (50 nM) for 12 h, followed by cDDP treatment for 12 h. (**G**) Representative PLA images show the interaction between COX-2 and p-Drp1^Ser616^ (Left). Scale bar, 20 μm. Quantification of COX-2/p-Drp1^Ser616^ PLA signals per cell is shown in the bar graph (Right). Data are expressed as the mean ± SD. * *p* < 0.05, compared to the corresponding control groups. ^#^ *p* < 0.05, compared to the corresponding siNC groups. (**H**) Cytoplasmic (Cyto) and mitochondrial (Mito) fractions were prepared and subjected to western blotting (Upper and Middle). Mitochondrial fractions immunoprecipitated with anti-COX-2 antibody were subjected to detection of the level and proportion of COX-2 and p-Drp1^Ser616^ (Lower). (**I**) Levels of apoptosis-related proteins Bax, cleaved Caspase-3, and Bcl-2 were determined by western blotting. Full Western Blot images can be found in Appendix A.

**Figure 8 cancers-14-00821-f008:**
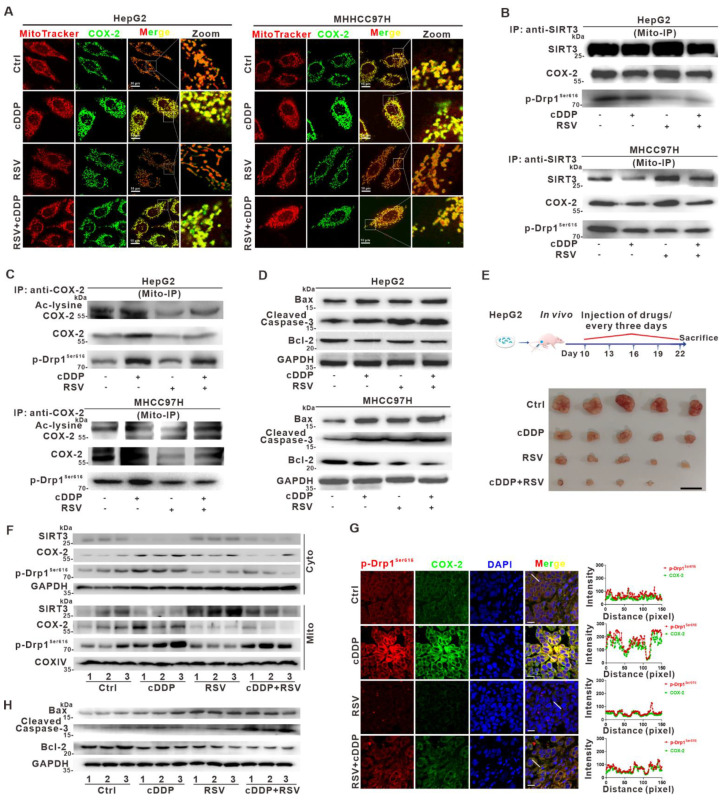
Deacetylation of mito-COX-2 via SIRT3 activation mediates the sensitivity of HCC to cisplatin by inhibiting mito-COX-2/p-Drp1^Ser616^ interaction in vitro and in vivo. (**A**–**D**) In our in vitro study, HepG2 and MHCC97H cells were pre-treated with or without RSV (50 μM) for 24 h, followed by cDDP treatment for 12 h. (**A**) Subcellular localization of COX-2 in HCC cells was examined by confocal microscopy. Representative immunofluorescence images of mitochondria (red) and COX-2 (green) were captured. Co-localization of COX-2 at mitochondria was examined. Scale bars, 10 μm. (**B**,**C**) Mitochondrial fractions were extracted to perform a mito-IP assay. Levels of SIRT3, COX-2 or ac-lysine COX-2, and p-Drp1^Ser616^ in mitochondrial complexes immunoprecipitated with indicated antibodies were determined. (**D**) The levels of apoptosis-related proteins Bax, cleaved Caspase-3, and Bcl-2 were determined by western blotting. (**E**–**H**) In our in vivo study, BALB/c nude mice bearing HepG2 cells as primary xenografts were randomly assigned to four groups (*n* = 5 for each group) at day 10. Subcutaneous injection was conducted for treatment with normal saline (Ctrl), cDDP (1 mg/kg), RSV (50 mg/kg), or a combination of cDDP and RSV every three days, four times. (**E**) The overall diagram of the in vivo study design (Upper). Images show the size of xenograft tumors (Lower). (**F**) Cytoplasmic (Cyto) and mitochondrial (Mito) fractions were prepared from xenograft tumor tissues and subjected to western blotting. Levels of SIRT3, COX-2, and p-Drp1^Ser616^ proteins in representative tumor tissues are shown. (**G**) Serial section of xenograft tumors was subjected to immunofluorescence (IF) assay to evaluate the expression of COX-2 (green) and p-Drp1^Ser616^ (red), while the co-localization of COX-2 and p-Drp1^Ser616^ was revealed in Merge (Left). Fluorescence curves were generated using Zen 2010 software (Right). Scale bars, 10 μm. (**H**) Levels of apoptosis-related proteins Bax, cleaved Caspase-3, and Bcl-2 in representative xenograft tumor tissues were determined by western blotting. Full Western Blot images can be found in Appendix A.

## Data Availability

The datasets supporting the conclusions of this article are included within the article.

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
