# Peer review of "Targeting Mitochondrial COX-2 Enhances Chemosensitivity via Drp1-Dependent Remodeling of Mitochondrial Dynamics in Hepatocellular Carcinoma"

_cancers, 2022, doi:10.3390/cancers14030821_

Round 1
Reviewer 1 Report
This is an interesting paper linking susceptibility of hepatocellular carcinomas and the COX2/Drp1-dependent signalling with susceptibility to anti-cancer drugs.
There are only a few minor points to look at. It is the usage of English. While generally pretty much OK, there are places where it should be polished up. Therefore, reading by a native speaker with some knowledge of biology would be useful.
In Figure 8E the authors show tumour growth documented by the volume of tumours at end-point. There should be complete growth curves, from the beginning of the experiment, for all 4 conditions.
In Figure 8I the authors present a cartoon attempting to summarise the major findings of the paper. This is rather confusing and hard to read. It is unclear to me what is on the left side (in red) and the right side (in blue). Also, why do the authors mention tumorigenesis here, which I understand as a malignant switch from normal liver tissue to malignant tissue - this is not included in the research documented in the manuscript. Of course, it would be interesting to see the level of COX2 during the malignant switch, but this is not the focus of the paper. The cartoon ought to be modified accordingly, such that it shows what the authors actually report on.
Reviewer 2 Report
In this study of Lin Che et al, authors have demonstrated that mitochondrial COX-2 is an important regulator of mitochondrial fission through its interaction with DRP1/PINK1-dependent manner. This function of COX-2 is also associated to several carcinogenis processess in HCC. The manuscript is well written and organized. The topic of the manuscript is interesting although the novelty of the manuscript is partially damaged by other studies reporting the link between COX-2, mitochondrial fission, DRP1 in non-liver cancers (e.g., 28435473). Additional improvements are required to support the conclusions raised by the authors. It is therefore recommended that authors follow these specific comments:
Major comments:
1- Authors used some GEO datasets to analyze the expression of COX-2, DRP1 etc in HCC patients. However, it is totally unclear why authors have chosen these specific datasets, which are limited in the number of patients. Moreover, GSE104310 display a lot of heterogeneity and a small number of patients. Therefore, the GSEA analysis of Figure 1G should also be repeated with other datasets with a higher number of patients. Authors should repeat their analyses (expression, correlations) in several other datasets, as well as in the TCGA cohort. Is there any difference with the etiologies? Among the patients analyzed by the authors, several are HBV positive. More information about patients should be incorporated in supplementary material. Authors should focus on one kind of hepatic carcinogenesis. In addition, one GEO dataset (GSE49515) is not HCC. The samples correspond to PBMCs from HCC patients. Another dataset should be used.
2-COX-2 is an inducible enzyme, whose expression is induced by pro-inflammatory cytokines (NFkB signaling). Beside COX-2 is a well-known tumor promoter, partially due t its catalytic activity involved in prostaglandins biosynthesis. What is the proportion of COX-2 translocating to the mitochondria in cells exposed to pro-inflammatory cytokines? Does inflammation prevent the interaction with DRP1 and thus its function on mitochondrial fission? This question can be answer by using cells where COX-2 expression is still inducible.
3- Are the kinases regulating pDRP1 (ser616) induced in HCC patients?
4-Authors made several comparisons between LO2 and HCC cells. However, the LO2 cells are immortalized and may not reflect completely a “normal hepatocyte”. Additional expereiments should be repeated using primary hepatocytes. Moreover, the comparisons shown should be analyzed in HCC patients vs peritumoral tissues (which is not really a normal liver).
5- A pharmacologic approach was used to activate PINK1. A genetic approach should also be used.
6- It is surprising that authors did not use a IgG control for all the immunoprecipitation experiments. This control must be provided to ensure the specificity of the results.
7- It is unclear why authors have used resveratrol to target SIRT3. This approach is not specific at all. Other more specific methods should be employed (siRNAs…).
8- All the western blots must be quantified and represented graphically with statistics (Mean +/- SD). Moreover, the whole membranes should be shown in supplementary materials, as required by the journal:
« Articles: Original research manuscripts. The journal considers all original research manuscripts, provided that the work reports scientifically sound experiments and provides a substantial amount of new information, e.g., research articles using only one cell line for the experiments will not be considered for publication (experiments need to be repeated on 1-2 more cell lines); authors should consider in vivo studies using orthotopic or transgenic models to validate gene function; for all Western blot figures, densitometry readings/intensity ratio of each band should be included; the whole Western blot showing all bands and molecular weight markers should be included in the Supplementary Materials; gene silencing experiments should use at least two gene-specific siRNAs, etc. »
In the supplementary file provided by the authors, it is not clear whether the membranes are cut or not and the molecular weight is missing.
Minor comments:
1-In the introduction, some info should be corrected. Currently, HCC is the seventh cancer and the second cause of cancer mortality
2- Authors should measure the expression of the endogenous COX-2 when overexpressing the MTS-COX-2. This will ensure that the effect of MTS COX-2 in the xenograft assays is not related to an increase of the endogenous COX-2, which can exert a tumor promoting function.
3-In figure 6E (Annexin V/PI staining), the compensation is not optimal.
Reviewer 3 Report
In this submission, Che et al. evaluated the underlying mechanism of mito-COX-2 and p-Drp1Ser616 interaction regulates chemotherapeutic response via mitochondrial dynamics in vitro and in vivo. This is an interesting study but a few mechanisms are not clear in its current format. I recommend major revisions before this paper can be accepted, which I have listed below;
- the key role of mitochondria in cancer progression has not been defined, a general role of enzymes can be discussed here too before discussing the specific enzymes. It will be helpful for readers to understand the mechanisms - second paragraph starts with mitochondria but herein I suggest adding a key mechanistic role of mitochondria in cancer make-up and therapeutics, authors should cite the paper; https://doi.org/10.1016/j.actbio.2021.04.054
- Line 90 - 'Mitochondrial COX-2 (mito-COX-2) has been recognized as a potential theranostic target against cancer stem cells in nasopharyngeal carcinoma'. this sentence certainly needs a reference. and also how this can be used as unit for both diagnostics and therapeutics simultaneously.
- Line 109-10: 'A growing body of evidence shows that SIRT3 inhibits tumorigenesis by deacetylation of its substrates in HCC mitochondria, pointing SIRT3 as a potential therapeutic target'. This needs more insightful information in terms of SIRT3 as a potential therapeutic target.
- Scale bar should be added to Figure 1 a and b histology images.
- Line 390" authors have repeatedly stated this 'Mitochondria are highly dynamic organelles that continuously undergo fission and fusion'. please avoid repeating the basic information on this.
- Platinum drugs should be introduced in 'introduction' section too. For example, three commonly used platinum chemotherapy drugs, cisplatin (cDDP), car- 624 boplatin (CBP), and oxaliplatin (L-OHP) which have been used in this work. Their mechanisms of action in terms of the impact of targeted mito-COX-2 intervention toward mitochondrial dynamics on the platinum-based chemotherapeutics in cells and in mouse model should be explained in further detail.
Round 2
Reviewer 2 Report
The manuscript has been improved and authors answered properly to my questions/comments
Reviewer 3 Report
I am pleased to recommend the revised manuscript for publication in Cancers.